# Body ownership promotes visual awareness

**Björn van der Hoort\*, Maria Reingardt, H Henrik Ehrsson**

Department of Neuroscience, Karolinska Institutet, Stockholm, Sweden

**Abstract** The sense of ownership of one's body is important for survival, e.g., in defending the body against a threat. However, in addition to affecting behavior, it also affects perception of the world. In the case of visuospatial perception, it has been shown that the sense of ownership causes external space to be perceptually scaled according to the size of the body. Here, we investigated the effect of ownership on another fundamental aspect of visual perception: visual awareness. In two binocular rivalry experiments, we manipulated the sense of ownership of a stranger's hand through visuotactile stimulation while that hand was one of the rival stimuli. The results show that ownership, but not mere visuotactile stimulation, increases the dominance of the hand percept. This effect is due to a combination of longer perceptual dominance durations and shorter suppression durations. Together, these results suggest that the sense of body ownership promotes visual awareness.

DOI: https://doi.org/10.7554/eLife.26022.001

## Introduction

Upon opening one's eyes and looking down at one's body, one becomes visually aware of one's hands, legs, and chest without exerting any effort. In addition to visual awareness of the body, one also has the sensation that what one is seeing is one's own body, that is, one experiences a sense of body ownership (henceforth, 'body ownership'). The feeling of body ownership distinguishes one's own body from external objects; this distinction is critical, for example, when an external object threatens the body. Because one's own body can be assumed to be more relevant than any other object, it is reasonable to expect that visual awareness will be boosted when, all else being equal, body ownership is present. In this study, we test this prediction.

Normally, body ownership is not experienced for objects other than the body. However, in the classic rubber-hand illusion, participants do experience ownership of an external object, namely, a rubber hand that is in front of them (*Botvinick and Cohen, 1998*). In this illusion, the participant's real hand is lying next to the rubber hand but is occluded from sight. After synchronous brushing of the rubber hand and the participant's real hand for a short period, the participant reports a sensation of ownership of the rubber hand and perceives that tactile sensations originate from the rubber hand. Most research on body ownership has focused on the perceptual rules that determine the rubber-hand illusion and similar ownership illusions. For example, these studies have shown that for ownership of an artificial hand to be induced, the hand must be placed in the same orientation as the real hand, and visuotactile stimulation applied to the rubber hand must be synchronously applied in the same direction as that applied to the veridical hand (*Makin et al., 2008*). From this line of research, it is clear that the rubber-hand illusion is a multisensory illusion (*Ehrsson, 2012*; *Graziano and Botvinick, 2002*; *Tsakiris, 2010*) and that ownership of limbs (*Makin et al., 2008*) and full bodies (*Blanke et al., 2015*; *Ehrsson, 2012*; *Petkova and Ehrsson, 2008*) depends on the dynamic integration of spatially and temporally congruent visual, tactile and proprioceptive information from a space near the body.

**\*For correspondence:**
bjorn.van.der.hoort@ki.se

**Competing interests:** The authors declare that no competing interests exist.

Recent studies have also shown that body ownership plays an important role in visual perception more generally, affecting how one sees the external world beyond one's body (*Banakou et al., 2013*; *Haggard and Jundi, 2009*; *Linkenauger et al., 2013*; *van der Hoort et al., 2011*). In particular, studies have shown that body size serves as a ruler in the size perception of objects in the external world, but only in the presence of ownership (*van der Hoort and Ehrsson, 2014*, *van der Hoort and Ehrsson, 2016*; *van der Hoort et al., 2011*). When participants experience body ownership of a small doll, other objects appear larger and farther away; conversely, ownership of a large artificial body makes other objects appear smaller and closer. Crucially, when ownership is disrupted, these effects are greatly diminished, although the visual stimuli are identical in the ownership and no-ownership conditions. Thus, body ownership appears to have a direct effect on the *content* of conscious visual perception that cannot be explained by differences in visual input. However, can ownership also affect the very *presence* of conscious visual perception? Can ownership enhance visual awareness? In the current study, we used binocular rivalry to address this question.

Binocular rivalry is an often-used method of studying visual awareness (*Blake and Logothetis, 2002*; *Lumer et al., 1998*). In binocular rivalry, the two eyes are presented with different visual images that compete for visual awareness, thus resulting in alternating subjective percepts (*Levelt, 1965*). Visual parameters such as contrast, spatial frequency, color, and motion together determine the 'stimulus strength' of each of the rival stimuli (*Brascamp et al., 2015*; *Levelt, 1965*). Increasing the strength of both stimuli equally increases the switching rate during rivalry, whereas increasing the strength of only one of the rival stimuli increases the overall perceptual dominance of that stimulus (*Brascamp et al., 2015*; *Levelt, 1965*).

Here, we investigated whether ownership can increase the perceptual dominance of a hand image as if it were increasing the strength of that visual stimulus. We hypothesized that because one's own hand is a more relevant object for the brain to process, visual awareness of a hand for which ownership is experienced should be prioritized. In the current study, one of the rival images consisted of a hand being stroked with a spherical object, which was presented while the participant's veridical hand was touched synchronously with an identical object to elicit illusory body ownership (*Figure 1*). We hypothesized that the overall dominance of the hand image would increase through the feeling of ownership of the hand. To isolate the effect of body ownership from effects related to tactile stimulation, visual impression of a hand image, or visuotactile stimulation, we compared the illusion condition to unimodal visual and tactile stimulation conditions (Experiment 1) and a bimodal visuotactile stimulation condition in which the hand image was anatomically incongruent (Experiment 2), that is, control conditions in which ownership was eliminated or significantly reduced. The results of the two experiments supported our hypothesis and demonstrated that body ownership facilitates visual awareness of a hand in the binocular rivalry paradigm.

## Materials and methods

### Subjects

We tested a total of 60 healthy volunteers, who were equally divided between Experiment 1 (mean age = 24.6; 18 women) and Experiment 2 (mean age = 26.2, 13 women), and who were naive to the purpose of the study. Participants were recruited through advertisements on the campus of the Karolinska Institutet. They received a cinema voucher as compensation. All participants had normal or corrected-to-normal sight. The number of participants for each experiment (*N* = 30) was based on previous studies in which body ownership was manipulated to investigate its effect on another variable of interest and to allow for correlation analyses (*Gentile et al., 2013*; *Guterstam et al., 2011*; *Kalckert and Ehrsson, 2012*).

### Ethics statement

Each participant signed an informed consent form before the onset of the experiment. The Regional Ethical Review Board of Stockholm approved the experimental procedures.

### Apparatus and stimuli

Visual stimuli were created with Final Cut Pro X (Apple, California, USA) (*Figure 1*). The hand image was a photographed hand that was isolated using a green screen and had a homogeneous red

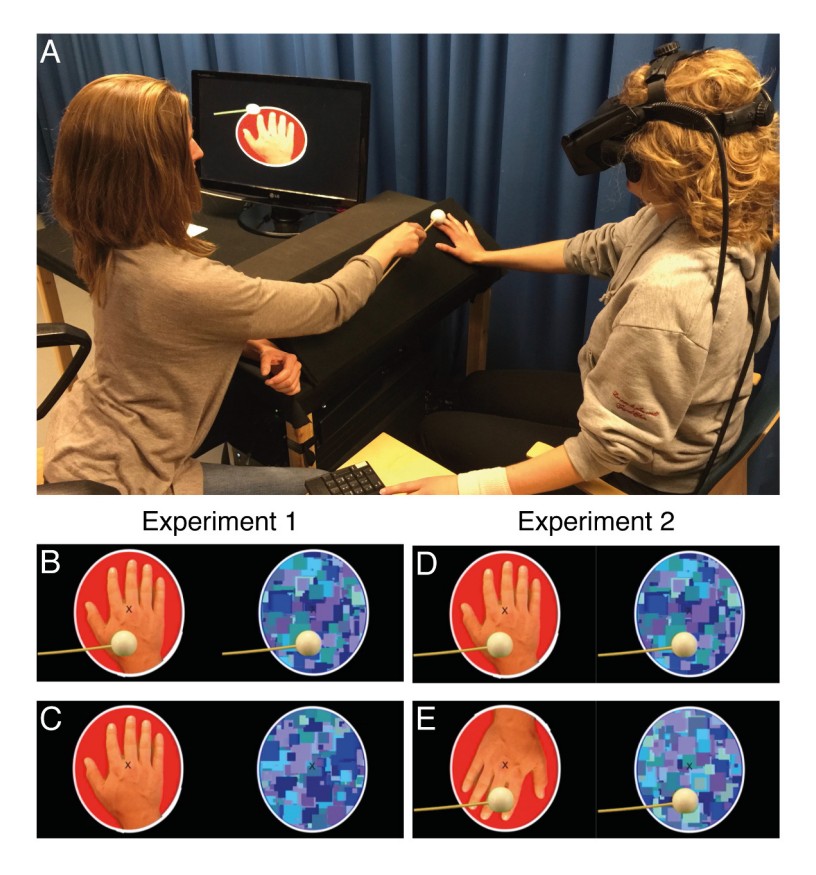

**Figure 1.** Experimental setup and visual stimuli. A participant indicates the current dominant percept with her left hand while the experimenter applies synchronous tactile stimulation to the right hand (**A**). The visual stimuli used in Experiment 1: the visuotactile condition and visual-only condition (**B**); the tactile-only condition, no-stimulation condition, and baseline condition (**C**); and the visual stimuli used in Experiment 2: the congruent visuotactile condition and congruent visual-only condition (**D**) and the incongruent visuotactile condition and incongruent visual-only condition (**E**).

DOI: https://doi.org/10.7554/eLife.26022.002

background added. Next, a video was made of the same hand being covered by a green screen (by folding the green screen back over the hand) while it was being touched with a Styrofoam sphere (37 mm in diameter) attached to a wooden stick (25 cm long, 4 mm in diameter) that approached the hand from the left side. This setup allowed the moving object to be isolated and presented in both the hand image and the mask image. Thus, in conditions in which the moving stick was present, it was present in both images and was therefore visible at all times.

The mask image consisted of 1000 Mondrian images, created with MATLAB, that randomly alternated at 10 Hz (*Stein et al., 2011*). The Mondrian images consisted of different-sized rectangles of different shades of blue. For all conditions, a fixation cross was present in the center of each image. The sizes of the visual stimuli, including the white outline, were approximately 10 degrees of the visual field, and the width of the hand was approximately 8 degrees. Half of the participants were shown the hand image displayed to their right eyes and the mask image displayed to their left eyes, and the other half of the participants were shown the reverse display.

Tactile stimulation of a participant's veridical hand consisted of single continuous strokes with the same object used to create the visual stimuli, from the back of the right hand to the tip of the right index finger (*Figure 1A*). The duration of each stroke was 0.67 s. These strokes were synchronous and spatially congruent with the moving stick from the visual stimuli. One stroke was applied every 1.5 s with the following pattern: three strokes (4.5 s), one rest (1.5 s), two strokes (3 s), one rest (1.5 s). During rest, the object was stationary next to the hand. In order to synchronize tactile stimulation

with the visual stimuli, the experimenter, wearing headphones (not shown in *Figure 1A*), listened to a click track that contained an audio-cue for each stroke.

## Task and procedure

Participants wore a set of head-mounted displays (HMDs; VR1280, Virtual Realities LLC, Texas, USA, with color displays showing 1280 × 1024 pixels with a 60-degree diagonal field of view) and placed their right hands on a diagonal platform on the table in front of them. They tilted their heads such that they would be looking straight at their own hand if they were not wearing the HMDs. Before the main experiment, they underwent a training session in which the distance between the displays could be adjusted to each participant's individual interocular distance. Participants fixated on the fixation cross at all times. Their task was to press and hold a button when they perceived the hand and to release the button when they perceived the mask. If they perceived a mix of both images, they were instructed to press the button only if more than half of their percept consisted of the hand image. Between blocks, participants had the option to take a short break.

Experiment 1 consisted of five conditions, with ownership of the hand being induced during one condition ('visuotactile') and ownership of the hand being absent or reduced during the other four conditions (see below). During the visuotactile condition, each participant's real hand was touched synchronously with the visual appearance of touch in the hand image to induce a sense of ownership of the hand image (*Figure 1B*). We included one condition ('tactile-only') to assess the isolated effect of tactile stimulation of each participant's real hand and one condition ('visual-only') to assess the isolated effect of the moving object in the visual stimuli (*Figure 1C*). In addition, we included a 'no-stimulation' condition to measure the rivalry dynamics of our visual stimuli in isolation, that is, in the absence of both the tactile component and the visual component of touch. Thus, tactile stimulation of a participant's veridical hand and the visual presence of the moving object could either be present or absent, thus yielding four conditions in a 2 × 2 design: visuotactile, visual-only, tactile-only, and no-stimulation. In all of these conditions, each participant's right hand was placed in a relaxed position on the platform in front of the participant, that is, anatomically and spatially congruent with the position of the hand in the image. On the basis of what is known from a large body of work on the rubber-hand illusion, we assumed that the visual-only and tactile-only conditions would not elicit a hand ownership illusion because they contain a mismatch between vision and touch (*Ehrsson, 2012*). The no-stimulation condition could theoretically induce a weak ownership illusion since there was less sensory mismatch between different modalities, but such a putative illusion would be significantly weaker compared to the visuotactile condition (*Guterstam et al., 2016*; *Samad et al., 2015*). In addition, we included a 'baseline' condition in which each participant's right hand was held in a relaxed position to the left side of the abdomen, that is, in a position that is incongruent with that of the hand in the image. This baseline condition allowed us to assess the effect of visuoproprioceptive hand congruency because it was similar to the no-stimulation condition in all other respects.

Each condition was tested in a separate block that lasted 270 s. The order of conditions was counterbalanced across participants. At the end of these five blocks, participants repeated the visuotactile condition for another 120 s, after which they filled out a questionnaire. The questionnaire contained four items: Q1 ('It felt as if the hand I saw was my own hand') and Q2 ('The touch I felt seemed to be caused by the white ball I saw') captured the illusion, whereas Q3 ('My own hand started to feel digital') and Q4 ('I felt as if I had two right hands at the same time') controlled for expectancy effects and task compliance.

Experiment 2 was designed to manipulate ownership by rotation of the hand image in two otherwise equivalent conditions, thereby isolating the effect of ownership from that of synchronous visuotactile stimulation itself. To this end, the one condition in which ownership was present was identical to the visuotactile condition from Experiment 1 ('congruent visuotactile') (*Figure 1D*). In the 'incongruent visuotactile' condition, the only difference was the orientation of the hand image, which was rotated 180 degrees (*Figure 1E*). The movement direction of the object that was touching the hand in the image was identical in retinal space to that in the congruent condition (but antidirectional in hand-centered space). In addition, two control conditions were included in which the hand image was presented in the two orientations to serve as a baseline for the visual stimuli in the absence of tactile stimulation. The moving object was still visible in these conditions, and the visual stimuli were therefore identical to those in the two visuotactile conditions. Thus, Experiment 2 had a 2 × 2 design in which congruency and tactile stimulation were manipulated. The hand image was congruent (as in

Experiment 1) or incongruent (rotated 180 degrees) to each participant's real hand, which received tactile stimulation (congruent visuotactile and incongruent visuotactile) or did not ('congruent visual-only' and 'incongruent visual-only'). As in Experiment 1, the duration of each condition was 270 s, and the order of conditions was semi-randomized. After the four blocks of the main experiment, participants repeated both visuotactile conditions (congruent and incongruent) and were given the same questionnaire as in Experiment 1.

## Data analysis and statistics

### Hand ownership illusion

Across different analyses, we performed Kolmogorov-Smirnov tests to assess the normality of the data. For the questionnaire data, we used Wilcoxon signed-rank tests because the data were measured on an ordinal Likert scale. In Experiment 1, only the visuotactile condition had a questionnaire included. Therefore, we compared the median scores for the illusion-related questions Q1 ('It felt as if the hand I saw was my own hand') and Q2 ('The touch I felt seemed to be caused by the white ball I saw') with those for control questions Q3 ('My own hand started to feel digital') and Q4 ('I felt as if I had two right hands at the same time') (see 'Task and procedure'). In addition, we calculated each participant's individual illusion score as (Q1 + Q2) – (Q3 + Q4). In Experiment 2, we included a questionnaire in both visuotactile conditions (congruent and incongruent), which allowed us to assess differences in each item of the questionnaire directly by using Wilcoxon signed-rank tests.

### Total dominance hand percept

The total time that the hand was seen was converted to a percentage of the total 270 s per condition. Next, we performed planned t-tests between conditions that used identical visual stimuli. Thus, for Experiment 1 we compared visuotactile versus visual-only; and tactile-only versus no-stimulation. Similarly, for Experiment 2 we compared congruent visuotactile versus congruent visual-only; and incongruent visuotactile versus incongruent visual-only. In addition, for Experiment 1, we performed a full factorial 2 × 2 repeated-measures ANOVA, with vision and touch being either present or absent, to assess the main effects of the visual component and the tactile component of a touch as well as their interaction effect. For Experiment 2, we performed a full-factorial 2 × 2 repeated-measures ANOVA with the factors of congruency (congruent versus incongruent) and tactile stimulation (visuotactile versus visual-only). Finally, we normalized the effect of the visuotactile condition to participants' overall mean ((visuotactile – mean)/mean) and correlated this perceptual effect with individual illusion scores (see previous paragraph) using a non-parametric Spearman correlation analysis.

### Switch rate

Next, we analyzed how often participants switched their percept from the hand to the mask and vice versa. This 'switch rate' is a measure of the overall perceptual stability and should be identical for conditions with identical visual stimuli and similar attentional demands. To test whether conditions with identical visual stimuli indeed had identical switch rates, we counted the total number of switches between the hand percept and the mask percept in each condition for every participant. This number was divided by the total duration of each block (270 s), thus yielding the switch rate per participant and per condition. We performed a 2 × 2 repeated-measures ANOVA as well as paired t-tests between conditions with identical visual input to analyze the effects of the visual and tactile components of touch on switch rates in Experiment 1 and the effects of congruency and tactile stimulation in Experiment 2.

### Duration of single percepts

The last step of the main analysis examined individual percept durations. To control for individual differences and improve statistical power, we normalized dominance durations to participants' mean percept duration. Thus, separately for hand percepts and mask percepts, we subtracted the participants' mean percept duration across all conditions from that of individual conditions and divided this difference by that same mean ((condition-mean – mean)/mean). We performed paired sample t-tests to analyze the difference between conditions with matched visual stimuli and used a 2 × 2 repeated-measures ANOVA to analyze the interaction effect across conditions.

To obtain a better estimate of the magnitude of the effect of visuotactile stimulation on percept durations, we collapsed the data from both experiments for the (congruent) visuotactile and (congruent) visual-only conditions (which were identical across experiments). The other conditions were not identical and therefore we re-normalized the data by dividing by the mean of only the visuotactile condition and the visual-only condition; ((visuotactile – visual-only)/mean) (mean = 0.5*(visuotactile +visual only)).

## The effect of individual touches on overall dominance

In addition to the analyses described above, all of which are common in binocular rivalry research, we performed some more advanced analyses that made use of the onset and duration of individual touches. For the first of these analyses, for each condition we plotted how dominance changed as a function of time around a single touch by flagging the onset of each touch and calculating the average dominance of the hand-percept between 667 ms before touch onset and 1334 ms after touch onset (see *Figure 7A* and *Figure 9A*). Next, we created the difference-waves between conditions that used identical visual stimuli by subtracting those conditions (Experiment 1: [visuotactile – visual-only] (see *Figure 7B*); and [tactile-only – no-stimulation] (see *Figure 7C*); Experiment 2: [congruent visuotactile – congruent visual-only] (see *Figure 9B*); and [incongruent visuotactile – incongruent visual-only] (see *Figure 9C*)). Finally, we created an interaction-wave for each experiment by subtracting the two difference-waves (Experiment 1: [(visuotactile – visual-only) – (tactile-only – no-stimulation)] (see *Figure 7D*); Experiment 2: [(congruent visuotactile – congruent visual-only) – (incongruent visuotactile – incongruent visual-only)] (see *Figure 9D*)). For the analysis, we averaged the dominance for each of six time bins, $T_{-1}$ (−667 to −333 ms), $T_0$ (−333–0 ms), $T_1$ (0–333 ms), $T_2$ (333–667 ms), $T_3$ (667–1000 ms), and $T_4$ (1000–1333 ms), to reduce multiple comparisons. The effect of time was assessed with a $6 \times 1$ repeated-measures ANOVA for each condition and for each of the difference and interaction waves. Additionally, we performed paired *t*-tests between identical time bins in different conditions. We applied Bonferroni corrections based on the number of time bins. To create the plots (*Figure 7* and *Figure 9*), we smoothed the data with a 33 ms moving average for clarity (analyses were performed on the raw data).

## The effect of individual touches on single percepts

A second analysis that made use of the onset of individual touches investigated whether changes in dominance durations and suppression durations were affected by these individual touches (*Lunghi et al., 2010*). We flagged the onset of the 667 ms pretouch phase ($T_{-1} – T_0$) and the onset of the 667 ms touch phase ($T_1 – T_2$). For each phase separately, we divided the data on the basis of whether the hand percept or the mask percept was dominant at the flag. For the resulting four groups of data (pretouch hand percept, pretouch mask percept, touch hand percept, and touch mask percept), we calculated how often the current percept was maintained, how often it had switched, and how often it had switched more than once. We ignored epochs in which the percept switched more than once. We analyzed the probability of maintaining a hand percept and the probability of switching when seeing a mask percept during both the pretouch phase and the touch phase. We performed a $2 \times 2$ repeated-measures ANOVA on each of those probabilities with the factors phase (pretouch and touch) and condition (visuotactile versus visual-only). Subsequently, for each phase, we performed paired *t*-tests between conditions with identical visual stimuli.

## Additional information

Effect sizes for Wilcoxon signed rank tests were calculated as $r = Z/\sqrt{N}$, where $N$ is the total sample size of a given test (i.e., $N = 2 \times 30 = 60$ for all tests) (*Fritz et al., 2012*). For paired *t*-tests, we calculated the effect size as Cohen's *d*, corrected for cross-condition correlations (*Morris and DeShon, 2002*). Effect sizes for independent *t*-tests were calculated as Cohen's *d*. For repeated-measures ANOVAs, we calculated the effect sizes of the main effects and interaction effect as partial eta squared ($\eta^2$). Finally, for Spearman correlations, the test statistic $\rho_S$ itself is the effect size. In addition to effect sizes, we report the 95% confidence interval (*CI*) for each statistical analysis. All statistical tests were two tailed, and no outliers were removed from analyses.

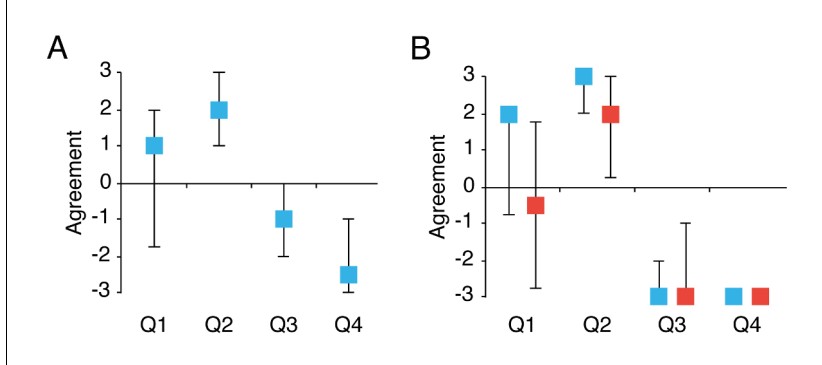

**Figure 2.** Ownership illusion. Median ratings of the different items of the questionnaire in Experiment 1 (**A**) and Experiment 2 (**B**). Q1: 'It felt as if the hand I saw was my own hand'; Q2: 'The touch I felt seemed to be caused by the white ball I saw'; Q3: 'My own hand started to feel digital'; Q4: 'I felt as if I had two right hands at the same time.' Q1 and Q2 capture the ownership illusion, and Q3 and Q4 control for expectancy effects and response bias. Blue indicates the congruent visuotactile condition; red indicates the incongruent visuotactile condition. Error bars indicate the interquartile range.
DOI: https://doi.org/10.7554/eLife.26022.003

# Results

## Experiment 1

### Hand ownership illusion

Nineteen participants out of the total group of 30 reported ratings of at least +1 on questionnaire statement Q1 ('It felt as if the hand I saw was my own hand'), and the median response was +1 (*Figure 2A*). In addition, 23 participants reported ratings of at least +1 on Q2 ('The touch I felt seemed to be caused by the white ball I saw'), and the median was +2. These proportions of participants affirming the illusion-related statements and the median scores are comparable to previous experiments with the rubber hand illusion (*Kalckert and Ehrsson, 2014*). The median rating of the control statements was lower than zero (Q3: *Mdn* = −1; Q4: *Mdn* = −2.5). Crucially, participants rated the illusion statements significantly higher than the control statements (*Z* = 4.44, *p* <0.001, *r* = 0.57). Thus, on a group level, these questionnaire results suggest ownership of the hand was successfully induced during the visuotactile condition.

### Total dominance hand percept

During the visuotactile condition, in which participants experienced ownership of the hand, participants observed the hand 43.6% of the time, which was more than in any other condition (baseline = 33.9%, no-stimulation = 34.5%, tactile-only = 36.5%, visual-only = 39.1%). Planned comparison revealed that the overall dominance of the hand was significantly larger in the visuotactile condition compared to the visual only condition (*mean difference ± SEM* = 4.48 ± 1.76, *CI* = 0.87–8.08, *t*(29) = 2.54, *p* = 0.017, *d* = 0.47). Additional post hoc tests showed significantly larger dominance in the visuotactile condition compared to the three other conditions as well (visuotactile vs baseline: *mean difference ± SEM* = 9.70 ± 2.36, *CI* = 4.88–14.53, *t*(29) = 4.11, *p* = 0.003, *d* = 0.75; vs no-stimulation: *mean difference ± SEM* = 9.10 ± 1.99, *CI* = 5.03–13.16 *d* = 0.85, *t*(29) = 4.58, *p* = 0.002; vs tactile-only: *mean difference ± SEM* = 7.07 ± 1.61, *CI* = 3.78–10.36, *t*(29) = 4.40, *p* <0.001, *d* = 0.80) (*Figure 3A*). Interestingly, the hand percept was not more dominant in the tactile-only condition than in the no-stimulation condition as revealed by our planned comparison (*mean difference ± SEM* = 2.02 ± 2.04, *CI* = −2.15–6.19, *t*(29) = 1.30, *p* = 0.204, *d* = 0.18). This result suggests that it is ownership of the hand, rather than mere tactile stimulation, that increases hand-percept dominance. However, the interaction between the visual and tactile components was not significant (*F*(1,29) = 0.99, *p* = 0.329, $\eta^2$ = 0.03; main effect of visual component: *F*(1,29) = 27.12, *p* <0.001, $\eta^2$ = 0.48; main effect of tactile component: *F*(1,29) = 5.03, *p* = 0.033, $\eta^2$ = 0.15). A final planned comparison was conducted between the no-stimulation condition, in which each

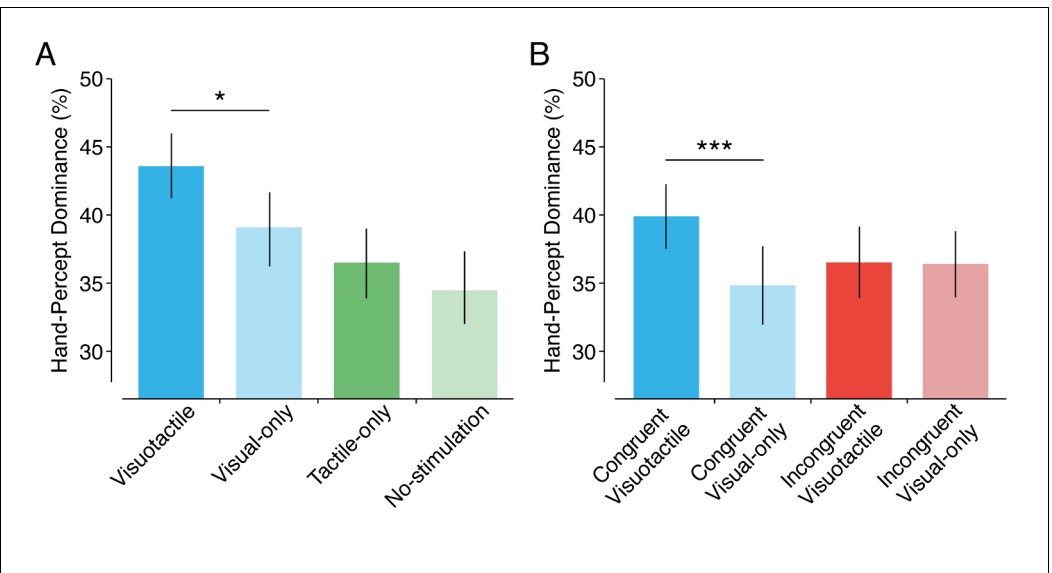

**Figure 3.** Hand-percept dominance. Average dominance of the hand percept in the different conditions of Experiment 1 (**A**) and Experiment 2 (**B**). Error bars indicate SE. *$p$ <0.05, ***$p$ <0.001.
DOI: https://doi.org/10.7554/eLife.26022.004

participant's real hand was in the same spatial location as that of the observed hand image, and the baseline condition, in which the participant's real hand was retracted next to the body in an incongruent position. The dominance of the hand percept was similar in both conditions (*mean difference ± SEM* = 0.61 ± 2.22, *CI* = −3.94–5.15, *t*(29) = 0.27, *p* > 0.25, *d* = 0.05), thus indicating that the mere congruent position of a participant's veridical hand did not increase the dominance of the hand image.

In a post hoc correlation analysis, we found that participants with higher ownership illusion scores had stronger effects on hand dominance in the visuotactile condition ($\rho_S$ = 0.53, *p* = 0.003) (*Figure 4A*). This finding suggests that the resulting ownership of the observed hand, rather than mere visuotactile stimulation, caused the increase in the dominance of the hand image.

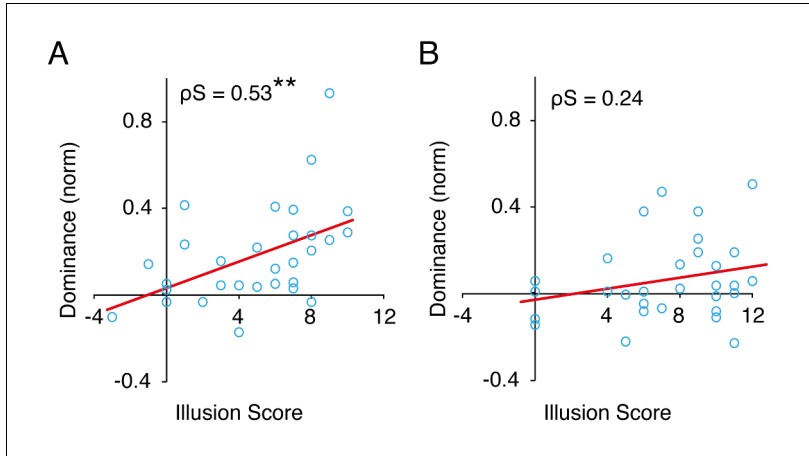

**Figure 4.** Ownership and dominance. Correlation between illusion scores (Q1 +Q2 - Q3 - Q4) and the normalized overall dominance in the visuotactile condition for Experiment 1 (**A**) and Experiment 2 (**B**). **$p$ <0.01.
DOI: https://doi.org/10.7554/eLife.26022.005

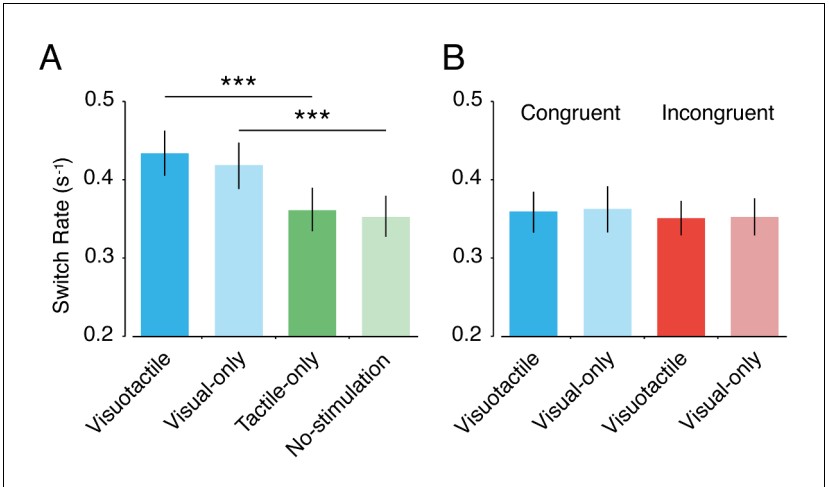

**Figure 5.** Switch rate. The mean switch rate in the different conditions of Experiment 1 (A) and Experiment 2 (B). Error bars indicate SEM. ***$p < 0.001$.

DOI: https://doi.org/10.7554/eLife.26022.006

## Switch rate

The presence of the moving object in the visuotactile and visual-only conditions changed the overall switching rate, that is, we found a main effect of the visual component of touch on the switch rate ($F$(1,29) = 39.97, $p < 0.001$, $\eta^2 = 0.58$) (*Figure 5A*). However, we did not find a main effect for the tactile component ($F$(1,29) = 1.129, $p = 0.297$, $\eta^2 = 0.04$) or an interaction effect ($F$(1,29) = 0.26, $p = 0.616$, $\eta^2 = 0.01$). The mean switch rate was higher for visuotactile (*mean* = 0.443 s$^{-1}$) than for tactile-only (0.367 s$^{-1}$) (*mean difference ± SEM* = 0.075 ± 0.015, *CI* = 0.045–0.106, $t$(29) = 5.11, $p < 0.001$, $d = 0.94$) and was higher for visual-only (0.424 s$^{-1}$) than for no-stimulation (0.360 s$^{-1}$) (*mean difference ± SEM* = 0.065 ± 0.016, *CI* = 0.032–0.097, $t$(29) = 4.10, $p < 0.001$, $d = 0.76$), but we found no differences between conditions with matched visual stimuli (visuotactile versus visual-only: *mean difference ± SEM* = 0.018 ± 0.015, *CI* = 0.045–0.106, $t$(29) = 1.25, $p = 0.221$, $d = 0.23$; tactile-only versus no-stimulation: *mean difference ± SEM* = 0.001 ± 0.018, *CI* = −0.029–0.044, $t$(29) = 0.435, $p = 0.667$, $d = 0.10$). Because conditions with different visual stimuli had different switch rates, it is difficult to interpret differences in percept durations between these conditions, that is, to dissociate overall perceptual stability (switch rate) from duration changes specific to one of the rival images. Therefore, the main focus of our individual percept duration analysis (see next paragraph) was on conditions with identical visual stimuli.

## Duration of single percepts

After normalizing the mean durations, we found that the visuotactile condition tended to increase the durations of the hand percept compared with the visual-only condition (*mean difference ± SEM* = 0.092 ± 0.046, *CI* = −0.003–0.044, $t$(29) = 1.98, $p = 0.057$, $d = 0.36$) (*Figure 6A*). In addition, the durations of suppression were significantly shorter in the visuotactile condition than in the visual-only condition (*mean difference ± SEM* = −1.01 ± 0.046, *CI* = −0.194 − −0.007, $t$(29) = −2.21, $p = 0.035$, $d = 0.40$). Importantly, these differences were not observed when the tactile-only condition was compared with the no-stimulation condition (hand percepts: *mean difference ± SEM* = 0.063 ± 0.066, *CI* = −0.072–0.199, $t$(29) = 0.96, $p = 0.347$, $d = 0.18$; mask percepts: *mean difference ± SEM* = −0.038 ± 0.081, *CI* = −0.203–0.127, $t$(29) = −0.47, $p = 0.641$, $d = 0.10$). Thus, the increased overall dominance of the hand image appears to be driven by a combination of prolonged hand percepts and shortened mask percepts. Furthermore, these changes in percept duration appear to be specific to visuotactile stimulation because tactile stimulation without a congruent visual component of touch did not significantly change percept durations compared with those in the no-stimulation condition. However, for both percepts, the interaction effect between the visual and tactile components was not significant (hand percepts: $F$(1,29) = 0.15, $p = 0.700$, $\eta^2 = 0.01$; mask percepts: $F$(1,29) = 0.50, $p = 0.485$, $\eta^2 = 0.02$).

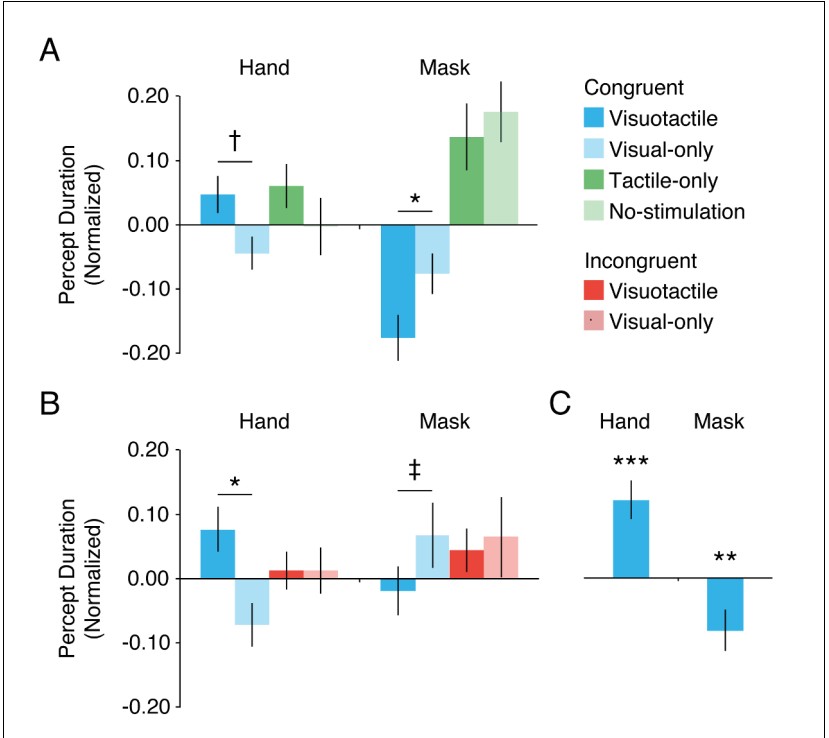

**Figure 6.** Percept durations. The mean normalized durations of the hand percept and the mask percept for Experiment 1 (A) and Experiment 2 (B) and the duration difference between the visuotactile condition and the visual-only condition across experiments (C). † $p = 0.057$, ‡ $p = 0.063$, *$p < 0.05$, ***$p < 0.001$.
DOI: https://doi.org/10.7554/eLife.26022.007

## The effect of individual touches on overall dominance

Next, we analyzed how dominance of the hand image changed as a result of an individual touch (*Figure 7A*). For the visuotactile condition, there was a strong effect of time regarding individual touches ($F(5,145) = 35.46$, $p < 0.001$, $\eta^2 = 0.55$). Hand dominance was significantly higher from the second half of a touch (T$_2$) onward than it was before a touch (T$_0$) (see *Table 1*). A similar effect of time was present for the visual-only condition ($F(5,145) = 56.89$, $p < 0.001$, $\eta^2 = 0.66$), and dominance was also significantly higher from T$_2$ onward (see *Table 1*). Surprisingly, individual touches did not affect dominance in the tactile-only condition ($F(5,145) = 1.40$, $p = 0.227$, $\eta^2 = 0.05$). Thus, the onset of tactile stimulation did not significantly increase hand dominance. Together, these results indicate that the visual impression of a touch causes a transient dominance increase of the hand image but that this effect is independent of ownership of that hand.

Next, we analyzed how the increased dominance in the visuotactile condition compared with the visual-only condition varied as a function of individual touches. *Figure 7B* shows the difference wave for the visuotactile condition and the visual-only condition. There was no effect of time on the difference between the visuotactile condition and the visual-only condition ($F(5,145) = 1.50$, $p = 0.195$, $\eta^2 = 0.05$). Instead, their difference was greater than zero throughout (see *Figure 7B*). If anything, the difference between these conditions appears to decrease during each individual touch. Thus, the overall dominance increase of the hand image in the visuotactile condition did not appear to be caused by individual touches.

Finally, we analyzed how the interaction effect changed in response to a single touch. *Figure 7D* shows the difference wave for *Figure 7B* and *Figure 7C*. Positive values indicate that the effect of tactile stimulation is larger when an object moves synchronously across the hand in the image, that is, it depicts the superadditive effect of visuotactile stimulation. We found a small effect of time on this interaction ($F(5,145) = 2.31$, $p = 0.047$, $\eta^2 = 0.07$). However, instead of the interaction effect being enhanced by individual touches, it weakened upon single touches (see *Figure 7D*).

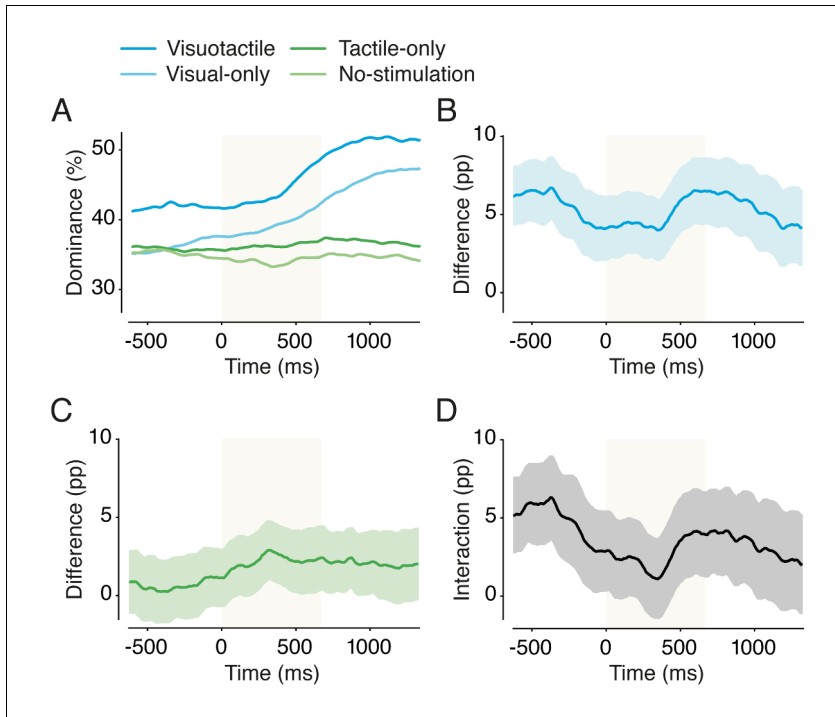

**Figure 7.** Effect of individual touches on hand-percept dominance (Experiment 1). Dominance of the hand percept as a function of time around individual touches (t = 0–667 ms) for the different conditions of Experiment 1 (**A**). The difference between the visuotactile condition and the visual-only condition (**B**), the difference between the tactile-only condition and the no-stimulation condition (**C**), and the difference between B and C (**D**). The gray zone indicates the period of touch (0–667 ms). For panels B-D: pp indicates percentage points (y-axis) and shading indicates SEM.

DOI: https://doi.org/10.7554/eLife.26022.008

This pattern of results is indicative of a continuous effect of ownership with an additional transient effect of single touches that is independent of ownership. To be more specific, these analyses show that visual information on the moving object causes a transient dominance increase for the hand image. However, this transient increase is not specific to the visuotactile condition, and single

**Table 1.** Paired *t*-test statistics.
Comparison of hand-image dominance between $T_0$ and $T_{1-4}$ (see **Figure 7** and **Figure 9**); *p*-values are Bonferroni corrected for multiple comparisons. Significant differences are displayed in bold.

| | | $T_1$ (0–333 ms) | $T_2$ (333–667 ms) | $T_3$ (667–1000 ms) | $T_4$ (1000–1334 ms) |
|---|---|---|---|---|---|
| Experiment 1 | Visuotactile | t(29) = 0.21 $p > 0.50$ | **t(29) = 3.49** $p = 0.008$ | **t(29) = 7.12** $p < 0.001$ | **t(29) = 7.72** $p < 0.001$ |
| | Visual-only | t(29) = 1.83 $p = 0.31$ | **t(29) = 4.68** $p < 0.001$ | **t(29) = 10.05** $p < 0.001$ | **t(29) = 9.58** $p < 0.001$ |
| Experiment 2 | Congruent Visuotactile | t(29) = 1.86 $p = 0.29$ | **t(29) = 3.85** $p = 0.002$ | **t(29) = 6.18** $p < 0.001$ | **t(29)=7.81** $p < 0.001$ |
| | Congruent Visual-only | t(29) = 1.02 $p > 0.50$ | **t(29) = 3.84** $p = 0.002$ | **t(29) = 6.75** $p < 0.001$ | **t(29) = 8.85** $p < 0.001$ |
| | Incongruent Visuotactile | t(29) = 1.15 $p > 0.50$ | **t(29) = 2.91** $p = 0.027$ | **t(29) = 5.15** $p < 0.001$ | **t(29) = 7.53** $p < 0.001$ |
| | Incongruent Visual-only | t(29) = 0.31 $p > 0.50$ | t(29) = 1.53 $p > 0.50$ | **t(29) = 5.11** $p < 0.001$ | **t(29) = 8.55** $p < 0.001$ |

DOI: https://doi.org/10.7554/eLife.26022.009

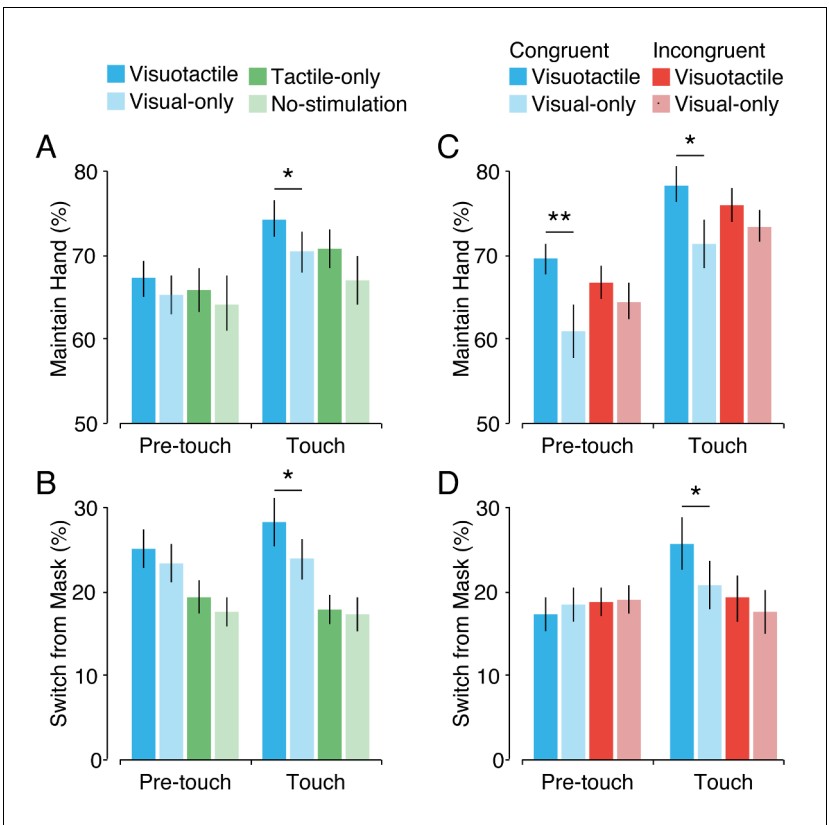

**Figure 8.** Maintaining and switching percepts. The probability of maintaining when the hand percept is dominant (**A, C**) and the probability of switching when mask percept is dominant (**B, D**), before a touch and during a touch, for Experiment 1 (**A, B**) and Experiment 2 (**C, D**). Error bars indicate SEM. *$p < 0.05$, **$p < 0.01$.
DOI: https://doi.org/10.7554/eLife.26022.010

touches therefore cannot explain the continuous increase in ownership elicited by the visuotactile stimulation. In contrast, after visual input and tactile stimulation are controlled for, the effect of the visuotactile condition seems largest before the onset of a touch, a result consistent with a continuing ownership effect. These findings provide additional evidence that ownership of a hand, a continuous experience, but not the visual or tactile component of touch alone, increases the overall dominance of that hand in the visuotactile condition.

## The effect of individual touches on single percepts

In addition to the effect of individual touches on overall dominance, we also analyzed how they affected single percepts. In a previous paragraph ('Duration of single percepts') we showed that hand percepts were maintained longer while mask percepts were shortened in the visuotactile condition compared with the visual-only condition. Here, we investigated whether these effects are driven by individual touches. To this end, we first calculated the probability of maintaining a hand percept during a touch, that is, when the visible stick was moving over the hand in the image ('touch' period: $T_1 + T_2$), and before the onset of a touch ('pretouch' period: $T_{-1} + T_0$), for the visuotactile and the visual-only condition (*Figure 8A*). The probability of maintaining a hand percept was higher overall during a touch than before a touch ($F(1,29) = 23.83$, $p < 0.001$, $\eta^2 = 0.45$). Furthermore, we found that the visuotactile condition had a tendency toward higher overall probabilities of maintaining a hand percept ($F(1,29) = 3.21$, $p = 0.084$, $\eta^2 = 0.10$). Although this effect seemed larger during a touch than before a touch (pretouch: *mean difference $\pm$ SEM* = 1.87 $\pm$ 2.34, *CI* = −2.93–6.66, $t(29)$ = 0.80, $p = 0.432$, $d = 0.15$; touch: *mean difference $\pm$ SEM* = 3.92 $\pm$ 1.66, *CI* = 0.53–7.32, $t(29)$ = 2.36, $p = 0.025$, $d = 0.44$), the interaction between phase and condition was not significant ($F(1,29)$ = 0.67, $p = 0.411$, $\eta^2 = 0.02$). Thus, maintaining the hand percept was more likely in the visuotactile

condition, but this effect did not rely on individual touches. When comparing the tactile-only condition with the no-stimulation condition, we found no main effect of condition ($F(1,29)$ = 4.93, $p$ = 0.160, $\eta^2$ = 0.07) and no interaction effect with individual touches ($F(1,29)$ = 0.53, $p$ = 0.472, $\eta^2$ = 0.02) (pretouch: *mean difference ± SEM* = 1.52 ± 2.39, CI = −3.37–6.41, $t(29)$ = 0.64, $p$ = 0.529, $d$ = 0.14; touch: *mean difference ± SEM* = 3.73 ± 2.34, CI = −1.06–8.52, $t(29)$ = 1.59, $p$ = 0.122, $d$ = 0.30).

In addition, we assessed how individual touches changed the likelihood of switching to dominance if the hand percept was suppressed (*Figure 8B*). We found that if the hand was suppressed, it was overall more likely to switch to dominance in the visuotactile condition than in the visual-only condition ($F(1,29)$ = 4.60, $p$ = 0.046, $\eta^2$ = 0.14). Across these conditions, switching was not more likely to occur during a touch ($F(1,29)$ = 1.89, $p$ = 0.181, $\eta^2$ = 0.06). Although the higher likelihood of switching in the visuotactile condition appeared to be more pronounced during individual touches (pretouch: *mean difference ± SEM* = 1.68 ± 1.71, CI = −1.83–5.19, $t(29)$ = 0.98, $p$ = 0.335, $d$ = 0.18; touch: *mean difference ± SEM* = 4.38 ± 2.14, CI = 0.01–8.75, $t(29)$ = 2.05, $p$ = 0.050, $d$ = 0.38), the interaction between phase and condition was not significant ($F(1,29)$ = 1.04, $p$ = 0.317, $\eta^2$ = 0.03). Thus, the likelihood of recovering a hand percept from suppression was greater in the visuotactile condition, and there was no significant change in this effect during touches. Furthermore, when comparing the tactile-only condition with the no-stimulation condition, we found no main effect of condition ($F(1,29)$ = 0.82, $p$ = 0.37, $\eta^2$ = 0.03) and no interaction effect ($F(1,29)$ = 0.52, $p$ = 0.45, $\eta^2$ = 0.02) with individual touches (pretouch: *mean difference ± SEM* = 1.91 ± 1.31, CI = −0.78–4.59, $t(29)$ = 1.45, $p$ = 0.157, $d$ = 0.27; touch: *mean difference ± SEM* = 0.55 ± 1.93, CI = −3.39–4.50, $t(29)$ = 0.29, $p$ = 0.78, $d$ = 0.05).

## Experiment 2

### Hand-ownership illusion

In the congruent visuotactile condition, 21 participants of the total group of 30 reported ratings of at least +1 on questionnaire statement Q1 ('It felt as if the hand I saw was my own hand'; *Mdn* =+2), and 26 participants reported ratings of at least +1 on Q2 ('The touch I felt seemed to be caused by the white ball I saw'; *Mdn* =+3) (*Figure 2B*). The ratings for Q1 and Q2 were significantly higher in the congruent condition than in the incongruent condition (Q1: $Z$ = 3.13, $p$ = 0.002, $r$ = 0.40; Q2: $Z$ = 2.36, $p$ = 0.018, $r$ = 0.30), whereas there were no differences for the control statements (Q3: $Z$ = −0.59, $p$ = 0.557, $r$ = 0.08; Q4: $Z$ = 1.73, $p$ = 0.173, $r$ = 0.22). In accordance with this, the illusion score was significantly higher in the congruent visuotactile condition than in the incongruent condition ($Z$ = 2.73, $p$ < 0.006, $r$ = 0.35). Thus, the experimental manipulation of ownership was successful.

### Overall dominance hand percept

Planned comparisons showed that the overall dominance of the hand image was higher in the congruent visuotactile condition (*mean* = 39.9%) than in the congruent visual-only condition (35.6%) (*mean difference ± SEM* = 4.28 ± 1.16, CI = 1.90–6.65, $t(29)$ = 3.68, $p$ < 0.001, $d$ = 0.68), but no such difference in hand dominance was observed for the incongruent conditions (visuotactile = 36.5%, visual-only = 36.4%, *mean difference* = 0.06 ± 1.40, CI = −2,92–2.80, $t(29)$ = 0.04, $p$ = 0.97, $d$ = 0.01) (*Figure 3B*). In fact, there was no main effect of congruency of the hand ($F(1,29)$ = 0.54, $p$ = 0.470, $\eta^2$ = 0.02), and the congruent visual-only condition did not differ from the incongruent visual-only condition in a post hoc comparison (*mean difference ± SEM* = 0.81 ± 2.11, CI = −5.11–3.50, $t(29)$ = 0.38, $p$ = 0.70, $d$ = 0.01) (*Figure 3B*). Thus, mere spatial congruency did not increase the dominance of the hand percept. Crucially, the interaction effect between visuotactile stimulation and congruency was significant ($F(1,29)$ = 6.63, $p$ = 0.015, $\eta^2$ = 0.19). This result indicates that visuotactile stimulation has an effect on dominance only if the hand is in a congruent position. Therefore, these results suggest that dominance is increased by the resulting body ownership and not by the visuotactile stimulation itself.

In a post hoc correlation analysis, we found that, in contrast to Experiment 1, the positive correlation between participants' ownership illusion scores and the increased dominance of the hand-image in the congruent visuotactile condition did not reach significance ($\rho_S$ = 0.24, $p$ = 0.197) (*Figure 4B*).

## Switch rate

In contrast to Experiment 1, the moving object (ball on a stick) was visible in all four conditions. However, the orientation of the hand image was different between the congruent conditions and the incongruent conditions. We found no effect of congruency on the switch rate ($F(1,29) = 0.62$, $p = 0.437$, $\eta^2 = 0.02$) (*Figure 5B*). In addition, as in Experiment 1, there was no main effect of tactile stimulation ($F(1,29) = 0.05$, $p = 0.83$, $\eta^2 = 0.00$) and no interaction effect ($F(1,29) = 0.02$, $p = 0.88$, $\eta^2 = 0.00$). As in Experiment 1, we found no differences between conditions with matched visual stimuli (visuotactile congruent versus visual-only congruent: *mean difference* $\pm$ *SEM* = 0.004 $\pm$ 0.013, CI = −0.022–0.029, $t(29) = 0.29$, $p = 0.771$, $d = 0.02$; visuotactile incongruent versus visual-only incongruent: *mean difference* $\pm$ *SEM* = 0.001 $\pm$ 0.018, CI = −0.030–0.032, $t(29) = 0.07$, $p = 0.947$, $d = 0.01$). Thus, participants had similar switch rates in all four conditions, indicating that the conditions were well matched in terms of perceptual stability of the visual stimuli.

## Duration of single percepts

As in Experiment 1, individual hand percepts lasted longer in the congruent visuotactile condition than in the congruent visual-only condition (*mean difference* $\pm$ *SEM* = 0.15 $\pm$ 0.04, CI = 0.07–0.22, $t(29) = 4.05$, $p < 0.001$, $d = 0.74$) (*Figure 6B*). In addition, suppression durations tended to be shorter in the visuotactile condition than in the congruent visual-only condition (*mean difference* $\pm$ *SEM* = −0.09 $\pm$ 0.04, CI = −0.18–0.00, $t(29) = −1.94$, $p = 0.063$, $d = 0.37$). Importantly, these differences were not observed between the incongruent conditions (hand percept: *mean difference* $\pm$ *SEM* = 0.00 $\pm$ 0.04, CI = −0.08–0.08, $t(29) = −0.01$, $p = 0.994$, $d = 0.00$; mask percept: *mean difference* $\pm$ *SEM* = −0.02 $\pm$ 0.08, CI = −0.18–0.14, $t(29) = −0.23$, $p = 0.803$, $d = 0.06$). In accordance with this finding, the interaction between congruency and visuotactile stimulation was significant for hand-dominance durations ($F(1,29) = 10.38$, $p = 0.003$, $\eta^2 = 0.26$), but this interaction was not significant for suppression durations ($F(1,29) = 0.56$, $p = 0.461$, $\eta^2 = 0.02$). Thus, only when the hand image was congruent with the participant's real hand did visuotactile stimulation lead to changes in percept durations. This result is consistent with our hypothesis that it is ownership of the hand, not synchronous visuotactile stimulation, that causes these changes in binocular rivalry dynamics.

The congruent conditions in Experiment 2 were identical to the visuotactile and visual-only conditions in Experiment 1 (the experiments differed only in their additional control conditions). Hence, we were able to collapse the data of both experiments for these two conditions to obtain more robust estimates of the magnitude of the effect of visuotactile stimulation, and the resulting presence of ownership, on percept durations. Across experiments, the congruent visuotactile condition, in which participants experienced ownership of the hand image, yielded longer hand-percept durations and shorter mask-percept durations than the congruent visual-only condition (hand percept: *mean difference* $\pm$ *SEM* = 0.14 $\pm$ 0.03, CI = 0.07–0.21, $t(59) = 4.06$, $p < 0.001$, $d = 0.62$; mask percept: *mean difference* $\pm$ *SEM* = 0.09 $\pm$ 0.04, CI = 0.02–0.17, $t(59) = −2.50$, $p = 0.015$, $d = 0.32$) (*Figure 6C*).

## The effect of individual touches on overall dominance

We assessed whether individual touches affected the dominance of the hand in different conditions (*Figure 9A*). As in Experiment 1, we found that individual touches affected the overall dominance of the hand image in the congruent conditions, that is, we found a main effect of time during the period surrounding a touch (visuotactile: $F(5,145) = 32.66$, $p < 0.001$, $\eta^2 = 0.53$; visual-only: $F(5,145) = 51.17$, $p < 0.001$, $\eta^2 = 0.64$). The onset of the dominance increase from $T_0$ was at $T_2$ (see *Table 1*), which is strikingly similar to the results of Experiment 1.

Next, we performed the same analysis for the incongruent conditions. If the dominance increase were truly due to the moving object, we would expect a similar effect in these conditions. Indeed, for both conditions, we found a strong effect of time during the period surrounding a touch (visuotactile: $F(5,145) = 27.71$, $p < 0.001$, $\eta^2 = 0.49$; visual-only: $F(5,145) = 34.75$, $p < 0.001$, $\eta^2 = 0.55$). The dominance increase from $T_0$ was first significant at $T_2$ in the incongruent visuotactile condition (see *Table 1*). Together, these results show that whenever there is movement of the stick, the probability of perceiving the hand image increases.

*Figure 9B and C* show the difference waves for the congruent and incongruent conditions, respectively. Positive values indicate more hand-percept dominance in the visuotactile condition.

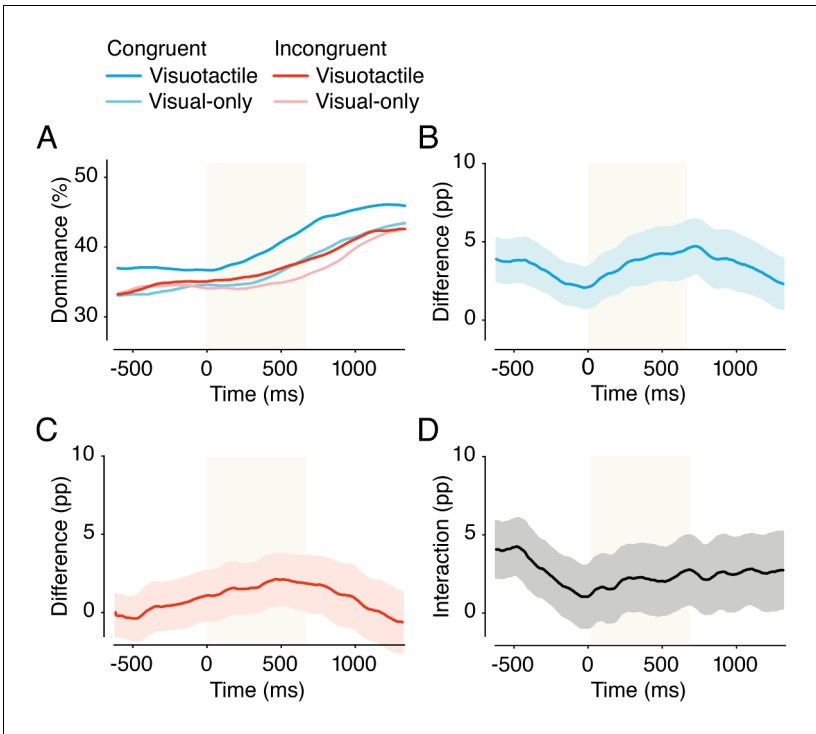

**Figure 9.** Effect of individual touches on hand-percept dominance (Experiment 2). Dominance of the hand percept as a function of time around individual touches (t = 0–667 ms) for the different conditions of Experiment 2 (A). The difference between the congruent conditions (B), the difference between the incongruent conditions (C), and the difference between B and C (D). The gray zone indicates the period of touch (0–667 ms). For panels B-D: pp indicates percentage points (y-axis) and shading indicates SEM.
DOI: https://doi.org/10.7554/eLife.26022.011

Individual touches did not affect the difference between visuotactile and visual-only for the congruent conditions ($F(5,145) = 0.86$, $p = 0.51$, $\eta^2 = 0.03$) (*Figure 9B*). The hand-percept dominance was significantly greater during the visuotactile condition than during the visual-only condition, before, during, and after a single touch. Thus, as in Experiment 1, the effect of visuotactile stimulation on hand dominance does not appear to be caused by individual touches. Instead, the results are more consistent with a continuous effect of ownership of the hand. The difference between the two incongruent conditions was also not affected by individual touches ($F(5,145) = 1.50$, $p = 0.193$, $\eta^2 = 0.05$) (*Figure 9C*).

Finally, we analyzed the interaction wave (*Figure 9D*), that is, the difference between the congruent difference wave and the incongruent difference wave, in which a positive value indicates a larger effect of visuotactile stimulation in the congruent conditions. Individual touches did not modulate this interaction ($F(5,145) = 0.923$, $p = 0.468$, $\eta^2 = 0.03$). As in Experiment 1, the interaction effect seems highest before the onset of touch and decreases during and after a touch (*Figure 9D*). In both experiments, the interaction effect seemed to decrease upon single touches, probably because the effects of seeing the moving object (and perhaps that of tactile stimulation) set in, reducing the relative effect of ownership. In the congruent visuotactile condition, the continuous effect that was independent of single touches was most likely caused by the continuous presence of ownership in these conditions, which was independent of single touches.

## The effect of individual touches on single percepts

For each of the four conditions in Experiment 2, we calculated the probability that a hand percept was maintained for the duration of a touch (667 ms). As a baseline, we also calculated that probability for the same time interval before the onset of a touch. First, we analyzed the difference between the congruent visuotactile condition and the congruent visual-only condition. We found that, overall,

maintaining a hand percept was more likely in the congruent visuotactile condition than in the congruent visual-only condition ($F_{(1,29)}$ = 12.05, $p$ = 0.002, $\eta^2$ = 0.29) (**Figure 8C**). In addition, across these two conditions, the probability of maintaining a hand percept was greater during a touch than before a touch ($F_{(1,29)}$ = 38.52, $p$ < 0.001, $\eta^2$ = 0.57). However, the difference in the probability of maintaining the hand percept between the visuotactile and visual-only congruent conditions did not change over time ($F_{(1,29)}$ = 0.26, $p$ = 0.618, $\eta^2$ = 0.01) (pretouch: *mean difference* ± *SEM* = 8.65 ± 2.49, *CI* = 3.55–13.74, $t_{(29)}$ = 3.47, $p$ = 0.002, $d$ = 0.74; touch: *mean difference* ± *SEM* = 7.12 ± 2.95, *CI* = 1.10–13.15, $t_{(29)}$ = 2.42, $p$ = 0.022, $d$ = 0.45). Thus, as in Experiment 1, the likelihood of maintaining the hand percept was higher in the congruent visuotactile condition than in the congruent visual-only condition, but this effect did not depend on single touches.

Next, for the same congruent visuotactile and visual-only conditions, we analyzed the probability of switching to the hand percept if the mask percept was perceived. This probability was higher overall during a touch than before a touch ($F_{(1,29)}$ = 11.52, $p$ = 0.002, $\eta^2$ = 0.28) (**Figure 8D**). Interestingly, although the overall higher switching probability in the congruent visuotactile condition did not reach significance ($F_{(1,29)}$ = 2.48, $p$ = 0.126, $\eta^2$ = 0.08), we found that the difference between the congruent visuotactile condition and the congruent visual-only condition was significantly greater during touch ($F_{(1,29)}$ = 4.34, $p$ = 0.046, $\eta^2$ = 0.13) (pretouch: *mean difference* ± *SEM* = 1.08 ± 1.65, *CI* = −4.45–2.30, $t_{(29)}$ = −0.56, $p$ = 0.520, $d$ = 0.12; touch: *mean difference* ± *SEM* = 2.63 ± 2.36, *CI* = 0.57–9.58, $t_{(29)}$ = 2.31, $p$ = 0.029, $d$ = 0.43). Thus, individual touches increased the likelihood of the hand percept breaking through suppression, but only in the visuotactile condition, in which ownership was present.

Importantly, these effects were not found in the incongruent conditions. The probability of maintaining a hand percept was not different between the incongruent visuotactile condition and the incongruent visual-only condition $F_{(1,29)}$ = 2.65, $p$ = 0.114, $\eta^2$ = 0.08) (pretouch: *mean difference* ± *SEM* = 2.31 ± 2.03, *CI* = −1.84–6.46, $t_{(29)}$ = 1.14, $p$ = 0.264, $d$ = 0.21; touch: *mean difference* ± *SEM* = 2.51 ± 1.53, *CI* = −0.63–5.65, $t_{(29)}$ = 1.64, $p$ = 0.113, $d$ = 0.30). In addition, although there was a higher overall probability of maintaining the hand percept during a touch ($F_{(1,29)}$ = 34.84, $p$ < 0.001, $\eta^2$ = 0.55), this effect of individual touches was similar for the visuotactile condition and the visual-only condition ($F_{(1,29)}$ = 0.10, $p$ = 0.922, $\eta^2$ = 0.00). Furthermore, there was no difference between these two conditions in the likelihood of switching from a mask percept to a hand percept ($F_{(1,29)}$ = 0.41, $p$ = 0.528, $\eta^2$ = 0.01) (pretouch: *mean difference* ± *SEM* = −0.29 ± 1.44, *CI* = −3.24–2.66, $t_{(29)}$ = −0.20, $p$ = 0.843, $d$ = 0.04; touch: *mean difference* ± *SEM* = 1.72 ± 1.41, *CI* = −1.17–4.60, $t_{(29)}$ = 1.22, $p$ = 0.233, $d$ = 0.23), and there was neither a main effect of single touches ($F_{(1,29)}$ = 0.10, $p$ = 0.755, $\eta^2$ = 0.00) nor an interaction effect ($F_{(1,29)}$ = 1.28, $p$ = 0.268, $\eta^2$ = 0.04).

Because the congruent conditions in Experiment 2 were identical to the visuotactile and visual-only conditions in Experiment 1, we collapsed the data of all 60 participants for those two conditions to improve statistical power and obtain a more accurate measure of the magnitude of the effect. Again, we found that the overall likelihood of maintaining a hand percept was greater in the visuotactile condition than in the visual-only condition ($F_{(1,59)}$ = 14.41, $p$ < 0.001, $\eta^2$ = 0.20). Furthermore, the likelihood of maintaining a hand percept increased during a touch ($F_{(1,59)}$ = 60.13, $p$ < 0.001, $\eta^2$ = 0.51), but this effect was similar for the visuotactile and visual-only conditions ($F_{(1,59)}$ = 0.02, $p$ = 0.893, $\eta^2$ = 0.00) (visuotactile: *mean difference* ± *SEM* = 7.94 ± 1.34, *CI* = 5.26–10.62, $t_{(59)}$ = 5.93, $p$ < 0.001, $d$ = 0.77; visual-only: *mean difference* ± *SEM* = 7.68 ± 1.46, *CI* = 4.76–10.60, $t_{(59)}$ = 5.26, $p$ < 0.001, $d$ = 0.68). Next, we analyzed the likelihood of recovering the hand percept from suppression for instances in which the mask was the dominant percept. Overall, hand percepts were more likely to break through suppression in the visuotactile condition than in the visual-only condition ($F_{(1,59)}$ = 7.10, $p$ = 0.010, $\eta^2$ = 0.11). Moreover, switching from the mask percept to the hand percept was more likely during a touch than before a touch ($F_{(1,59)}$ = 11.80, $p$ < 0.001, $\eta^2$ = 0.17). Importantly, this effect of single touches on the likelihood of switching was greater in the visuotactile condition than in the visual-only condition ($F_{(1,59)}$ = 5.00, $p$ = 0.029, $\eta^2$ = 0.08) (visuotactile: *mean difference* ± *SEM* = 5.77 ± 1.73, *CI* = 2.31–9.22, $t_{(59)}$ = 3.34, $p$ = 0.002, $d$ = 0.45; visual-only: *mean difference* ± *SEM* = 1.34 ± 1.06, *CI* = 0.78–3.46, $t_{(59)}$ = 1.27, $p$ = 0.21, $d$ = 0.17). Together, these analyses show that the longer maintenance of hand percepts was caused not by single touches but by a sustained effect of ownership of the hand. In contrast, recovering the hand from suppression does appear to rely on single touches, but only in the specific

condition in which ownership of the hand is experimentally induced, that is, the congruent visuotactile condition. Interestingly, this suggests that even though the hand percept is suppressed, some information about ownership of the hand is maintained.

## Discussion

In two separate binocular rivalry experiments, we experimentally manipulated the sense of ownership of a stranger's hand, which was one of the rival images. A key finding was that in the presence of body ownership, the overall perceptual dominance of that hand increased. Body ownership was induced when a participant received tactile stimulation of his or her own hand in a manner that was synchronous with and spatially congruent to the object that touched the hand in the image (the visuotactile condition). Interestingly, participants who reported stronger ownership also had a greater effect of visuotactile stimulation on the dominance of that hand image. Moreover, mere tactile stimulation of each participant's real hand without induction of ownership (tactile-only condition) did not increase the dominance of the hand image. Together, these results suggest that it is body ownership, rather than mere synchronous visuotactile stimulation, that increases the dominance of the hand image. To test this hypothesis directly, Experiment 2 compared synchronous visuotactile stimulation in the presence of ownership (i.e., when the hand image was spatially congruent with the real hand) with synchronous visuotactile stimulation in the absence of ownership (i.e., when the hand image was spatially incongruent). As expected, only when the hand image was spatially congruent and ownership was induced did visuotactile stimulation increase the dominance of the hand image. Crucially, in both experiments, we found that this dominance increase correlated positively with participants' individual ownership illusion score. Moreover, these effects were greatest before the onset of single touches, in agreement with the interpretation that it is sustained ownership of the hand that drives the dominance effect rather than transient visuotactile events. The finding that ownership promotes visual awareness provides an important new role for ownership in sensory processing and helps to explain the central role of body ownership in conscious awareness. We know from earlier studies that ownership can change the content of conscious perception in such ways as enhancing tactile acuity (*Haggard et al., 2003*; *Longo et al., 2008*), calibrating proprioception (*Abdulkarim and Ehrsson, 2016*; *Botvinick and Cohen, 1998*; *Tsakiris and Haggard, 2005*), and scaling visuospatial perception (*van der Hoort and Ehrsson, 2014*, *van der Hoort and Ehrsson, 2016*; *van der Hoort et al., 2011*). The fundamental advance made in this study is the demonstration that body ownership can affect the very presence of conscious perception, namely, visual awareness. How, exactly, does body ownership promote visual awareness in the binocular rivalry paradigm?

An increase in overall dominance can be caused by longer dominance durations or by shorter suppression durations. It is important to distinguish between the two because dominance and suppression rely on distinct neural processes (*Blake and Logothetis, 2002*). We found evidence for both: ownership increased the durations of individual hand percepts and decreased the durations of suppression. The hand percepts were maintained longer independently of individual visuotactile events. This effect cannot be explained by increased voluntary attention toward the hand image for two reasons. First, we did not find a difference in switch rate between conditions with identical visual input, suggesting that the overall attentional demand is the same between conditions (*Paffen et al., 2006*). Second, the effect of ownership on suppression durations, which is not affected by voluntary attention (*Ooi and He, 1999*; *Paffen and Alais, 2011*), was of similar magnitude to the effect of ownership on dominance durations. Instead, the prolonged dominance suggests that ownership effectively increases the 'stimulus strength' of the visual hand stimulus, similar to an increase in contrast (*Brascamp et al., 2015*), and thereby prevents the suppressed mask stimulus from entering the subject's awareness. In contrast to prolonged dominance, the decreased suppression durations seemed to be driven by single visuotactile events, but only when there was ownership of the suppressed hand. This result is intriguing because the hand is not perceived at the onset of touch. This result suggests that traces of the suppressed hand stimulus still contain information about ownership and that these traces are perceptually bound to the tactile stimulus in the case of ownership in order to break through suppression.

What might the neural mechanisms underlying these effects involve? Hand ownership is known to arise from multisensory integration in the intraparietal sulcus (IPS) and the premotor cortex

(*Ehrsson, 2012*; *Ehrsson et al., 2004*; *Gentile et al., 2013*). Visual perception of hand images occurs in a part of the lateral occipital complex (LOC) known as the extrastriate body area (EBA) (*Downing et al., 2001*), which is additionally able to dissociate between the visual impressions of one's own hand and someone else's hand (*Myers and Sowden, 2008*). During hand-ownership illusions, the LOC/EBA is activated (*Gentile et al., 2013*; *Limanowski et al., 2014*), and the effective connectivity between this area and the IPS is increased, indicating increases in neural communication between the two regions (*Gentile et al., 2013*; *Guterstam et al., 2013*). Therefore, we hypothesize that functional connectivity between the IPS and the EBA increases the strength of the visual hand stimulus, thereby resulting in longer dominance durations of that hand. During instances of suppression, traces of the suppressed hand image may travel along the dorsal stream and reach the IPS, as is the case for suppressed images of tools (*Fang and He, 2005*). Here, signals related to this suppressed hand image would integrate with spatiotemporally congruent signals associated with the tactile stimulus and consequently increase the signal-to-noise ratio of the hand image in the EBA, increase its inhibition of signals related to the dominant mask percept, and thereby speed up its recovery from suppression.

The above discussion brings us to the conceptual issue of whether body ownership boosts visual awareness 'directly' at the higher level of conscious selection, or if body ownership influences visual awareness 'indirectly' by first increasing the signal strength in lower level visual representations, which in turn leads to facilitation of awareness (*Giles et al., 2016*). Our results cannot distinguish between these two alternatives. We know from a recent study by *Zou et al. (2016)* that invisible stimulus features can induce binocular rivalry, which shows that the subjective perceptual alternation in this paradigm can be influenced by signal processing at early stages of visual processing. Body ownership illusions on the other hand are associated with multisensory integration in fronto-parietal association cortices (*Ehrsson et al., 2004*; *Gentile et al., 2013*; *Limanowski and Blankenburg, 2016*; *Petkova et al., 2011*; *Preston and Ehrsson, 2016*) – brain regions that are also implied in theories of consciousness such as the Global Work Space theory (*Dehaene and Naccache, 2001*; *Lau and Rosenthal, 2011*), so it is possible that ownership influences its effects at this higher level. The interpretations and speculations outlined in the preceding two paragraphs fit well with the explanation that body ownership could influence visual processing of the hand image in lower level representations by increasing its signal strength, which in turn leads to the enhanced perceptual dominance of the hand. However, it should be pointed out that this account also fits well with the recurrent processing theory of consciousness (*Lamme, 2006*), which claims that awareness is the result of recurrent processing. According to this theory body ownership could increase recurrent processing from the IPS to the EBA, and thereby 'directly' increase visual awareness of the hand image. Thus, although our results clearly show that body ownership increases the subjective dominance of the hand image, the possible existence of a direct causal relationship between body ownership and conscious awareness remains to be clarified in future studies.

Previous studies have extensively shown that the dynamics of binocular rivalry is affected by nonvisual stimuli that are congruent to one of the rival images. These nonvisual stimuli can be tactile (*Lunghi and Alais, 2013*; *Lunghi et al., 2010*), auditory (*Conrad et al., 2010*; *Sekuler et al., 1997*), and even olfactory (*Zhou et al., 2010*) in nature. These effects exemplify the role of multisensory processing in resolving perceptual ambiguity (*Ernst and Bülthoff, 2004*). However, although we used tactile stimulation to induce ownership, the current findings differ from these earlier findings in that rivalry dynamics were altered by ownership specifically and not tactile stimulation in general. Ownership itself does not belong to a single sensory modality but arises when visual, tactile, and proprioceptive information are congruent and can be perceptually bound. Another difference in the effects of tactile stimulation is that these effects have been attributed mainly to visuotactile interactions occurring at lower levels of visual processing (*Lunghi et al., 2010*), whereas the effect of ownership seems more likely to occur in the IPS and EBA (*Gentile et al., 2013*; *Guterstam et al., 2013*; *Limanowski et al., 2014*).

The ownership illusion used in this binocular study is different from the classic rubber-hand illusion (*Botvinick and Cohen, 1998*) in several aspects. First, the hand was presented to only one eye, and therefore perception of the hand was two-dimensional. Second, the image of the hand was considerably smaller than a life-sized rubber hand. The reason for this was that larger rival images tend to render more mixed-percept states, and we aimed to achieve rivalry dynamics that would make which percept was dominant less ambiguous to reduce response biases. Both the hand's small size

(*Pavani and Zampini, 2007*) and its two-dimensional nature (*IJsselsteijn et al., 2006*) may have caused the ownership illusion to be weakened compared with a classical rubber-hand illusion. However, the results of our questionnaire data demonstrated that we elicited a relatively strong limb-ownership illusion, with affirmative ratings of ownership that were comparable to those from previous studies with actual rubber hands (*Kalckert and Ehrsson, 2014*). Nevertheless, one can speculate that ownership of one's real hand should have an even larger effect on visual awareness than our experimental design allowed for.

In addition to the main results regarding the effects of ownership on dominance and suppression, our results show two other interesting phenomena. First, in each experiment, we included two conditions that differed only in terms of the visuoproprioceptive congruency between the participant's real hand and the hand image in the absence of any dynamic tactile stimulation, and we found that this methodology did not affect the rivalry dynamics of the hand image. In Experiment 1, we found that the rivalry dynamics was similar when participants positioned their hands in front of their bodies, in spatial alignment with the hand image (visual-only), and when participants folded their hands to the left side of their abdomens, in spatial incongruence with the hand image (baseline). In Experiment 2, we found that rotating the hand image itself, so that it was anatomically incongruent with the participant's hand in front of the body, did not affect rivalry dynamics in the absence of visuotactile stimulation. Thus, visuospatial congruency of the seen and felt position of the hand did not overcome suppression of that hand. This result seems to be in conflict with a recent continuous flash suppression experiment (*Salomon et al., 2013*) indicating that a hand image that was congruent with the participant's hand recovered from suppression slightly faster (60 ms) than a hand image that was incongruent to the participant's real hand. However, in the current study, we used binocular rivalry, which might be less sensitive to such small changes in suppression durations. A second interesting observation comes from the analysis of the visual-only conditions in which the moving object (ball on a stick) was visible at all times because it was part of both images. As soon as this object started to move, the dominance of the hand image increased. Because visuospatial context is known to affect binocular rivalry (*Alais and Blake, 1999*), this result indicates that the hand image provides a better context for the moving object than the mask image. To our knowledge, this is the first example of a contextual effect in binocular rivalry that is specific to movement. The hand image probably provides a more congruent visual context because it has similar depth cues (e.g., shadow and visual perspective) for the moving stick and because it is a more ecological visual stimulus than the mask image. We speculate that this finding relates to movement close to a hand being a strong cue for stimulation in peripersonal space, a stimulus that activates visuotactile neuronal populations in the fronto-parietal association cortices (*Brozzoli et al., 2011*; *Fogassi et al., 1996*; *Graziano et al., 1997*; *Makin et al., 2007*), including those with mirror-neuron properties (*Brozzoli et al., 2013*; *Ishida et al., 2010*).

Together, the results of this study suggest that ownership of a hand increases the visual awareness of that hand, both by increasing the strength of the hand-image, as if it increases in contrast, and by overcoming suppression when the hand is being touched. These findings increase the understanding of the role of ownership in visual perception in that ownership seems to enhance visual processing of a body part that is ours because it is more relevant to us, especially when it is being touched by an external source.

## Acknowledgements

This research was made possible by funding from the Swedish Research Council, the James McDonnell Foundation, Söderbergska Stiftelsen, and Riksbankens Jubileumsfond. We would like to thank Martti Mercurio for assistance with the technical aspects of the experimental setup.

## Additional information

### Funding

| Funder | Author |
| --- | --- |
| James McDonnell Foundation | H Henrik Ehrsson |

| Söderbergska Stiftelsen | H Henrik Ehrsson |
| --- | --- |
| Riksbankens Jubileumsfond | H Henrik Ehrsson |
| Swedish Research Council | H Henrik Ehrsson |

The funders had no role in study design, data collection and interpretation, or the decision to submitthe work for publication

## Author contributions

Björn van der Hoort, Conceptualization, Formal analysis, Supervision, Investigation, Visualization, Methodology, Writing—original draft, Writing—review and editing; Maria Reingardt, Data curation, Writing—review and editing; H Henrik Ehrsson, Conceptualization, Supervision, Funding acquisition, Project administration, Writing—review and editing

## Author ORCIDs

Björn van der Hoort http://orcid.org/0000-0002-2346-8373

## Ethics

Human subjects: Each participant signed an informed consent form before the onset of the experiment. The Regional Ethical Review Board of Stockholm approved the experimental procedures.

## Decision letter and Author response

Decision letter https://doi.org/10.7554/eLife.26022.013
Author response https://doi.org/10.7554/eLife.26022.014

## Additional files

### Supplementary files

• Transparent reporting form
DOI: https://doi.org/10.7554/eLife.26022.012

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
