## [Decision Letter]

Thank you for submitting your article "Body-ownership promotes visual awareness" for consideration by *eLife*. Your article has been reviewed by three peer reviewers, and the evaluation has been overseen by a Reviewing Editor and Sabine Kastner as the Senior Editor. The following individuals involved in review of your submission have agreed to reveal their identity: Megan AK Peters (Reviewer #2).

The reviewers have discussed the reviews with one another and the Reviewing Editor has drafted this decision to help you prepare a revised submission.

Summary:

The paper, "Body ownership promotes visual awareness" by van der Hoort et al. describes a psychophysical experiment in humans in which binocular rivalry was used to study visual awareness. In one of the rivalrous images, a hand was shown. Through the use of the rubber hand illusion, a hand was made to seem a part of the subject's body or not. When incorporated into the subject's own body, the visual hand was dominant in the binocular rivalry. The experiment suggests that some aspect of the rubber hand procedure interacts with visual awareness. Through several control conditions and a second experiment, the experimenters narrowed down the possibilities and suggest that specifically, the incorporation of a visual stimulus into the body schema causes it to increase in visual awareness.

Essential revisions:

The reviewers and editors were positive about the study. However, they identified a number of issues that will need to be considered in a revision, before the paper can be further considered for publication.i) You need to demonstrate that ownership was the critical factor that affected dominance. Please address the specific concerns raised by reviewer 3 with respect to experiment 1 and provide a correlation analysis between dominance and ownership for experiment 2.ii) Please address carefully the various statistical concerns that were mentioned in the reviews.iii) Please improve the overall clarity of presentation and writing.iv) Please tone down some of the claims, as outlined by all three reviewers.

Reviewer #1:

The paper, "Body ownership promotes visual awareness" by van der Hoort et al. describes a psychophysical experiment in humans in which binocular rivalry was used to study visual awareness. In one of the rivalrous images, a hand was shown. Through the use of the rubber hand illusion, at hand was made to seem a part of the subject's body or not. When incorporated into the subject's own body, the visual hand was dominant in the binocular rivalry. The experiment shows that some aspect of the rubber hand procedure interacts with visual awareness. Through several control conditions and a clever second experiment, the experimenters narrowed down the possibilities quite a bit. The experimenters suggest that specifically, the incorporation of a visual stimulus into the body schema causes it to increase in visual awareness. As long as this conclusion is presented in a cautious way, the paper seems like a useful and very interesting contribution to me. It is clearly written and the analysis is especially thorough. I have very little to suggest for improvement.

A few comments throughout could be toned down. For example, in the Introduction, "In the current study, we used binocular rivalry to answer this question." A lot of readers will take issue with this misplaced confidence. The study does not answer the question. It addresses the question; it provides evidence; some people will be convinced (especially by the very nice experiment 2) but others will need more data to convince them it isn't a different, correlated process that explains the results. For example, it is always possible that body ownership causes a slight increase in attention, and that the crucial factor is attention, however it is directed, whether by body ownership illusion or not.

The last paragraph of the paper also contains some overly confident sentences. It's very tricky to claim that you have absolutely shown a causal relationship. The results suggest it.

Reviewer #2:

van der Hoort and colleagues use a binocular rivalry task to measure the effect of body ownership on visual dominance times – a measure of conscious perceptions. They find that induction of the rubber hand illusion (i.e., induction of ownership over a rubber hand) is correlated with longer dominance times, across the same visuotactile stimulation conditions. They conclude that ownership promotes visual awareness.

Overall assessment and major comments:

I like the idea of this study. It is clearly motivated and generally well executed, and is presented clearly. I have some conceptual concerns about the interpretation of the results, however.

Personally, I'm not convinced that binocular rivalry measures visual awareness per se. Zou, He, and Zhang (2016, PNAS) showed that even unconsciously processed conflicting visual inputs can lead to conscious perceptual rivalry, which suggests that rivalry reflects a lower-level alternation or oscillation of ocular dominance rather than subjective awareness per se (see Giles, Lau, and Odegaard, 2016 for discussion). The same could theoretically be going on in the present study, in that the fluctuation in signal is occurring lower than the "visual awareness" stage. In the parlance of hierarchical causal inference in multisensory integration, perhaps the pre-conscious processing ends up selecting the visual input that is congruent with the posterior probability of a common cause that the system arrives at (see Samad, Chung, and Shams, 2015). That is, when the perceptual system assigns a high posterior probability of a common cause (high binding probability), the visual input that is congruent with that inference is boosted (via gain enhancement, or some other mechanism of signal strength enhancement) while the visual input that is incongruent with the inference (or irrelevant to it) is suppressed. This may also be compounded by the fact that binocular rivalry experiments kind of force the observer into a model selection or probability matching decision strategy, rather than a model averaging kind of framework (see Wozny et al., 2010 PLoS Comp Bio), because it's less easy to blend the inputs from the two eyes in a weighted averaging kind of way (although partial breakthrough does occur). So if this is what's happening, it's a low level selection of ocular dominance and signal strength boosting that may have little to do with awareness per se.

I don't think this conceptual distinction detrimentally impacts the general motivation for the paper, which the authors state right in the first paragraph of the Introduction as, "Because one's own body is more relevant than any other object, it is reasonable to expect that visual awareness will be boosted when, all else being equal, body-ownership is present." (Sidenote: could the authors provide some citation for the statement, 'Because one's own body is more relevant than any other object'?) But I think it does impact the use of the term – and concept – "visual awareness": the conclusions of the paper are also consistent with the idea that when body-ownership is present, a low level signal is itself boosted, which then causes longer dominance times. It's a sort of subtle difference, but it's really about where in the processing stream the effects of body-ownership will have their effect. Indeed, the authors mention in the Introduction that rivalrous dominance times can be increased simply by increasing the strength of one eye's visual input, and then again in the Discussion, and I think that explanation of an internal boost in signal strength due to congruency with the result of a causal inference process about ownership/binding might also apply to the results of this study. So unfortunately, I don't think the evidence here is strong enough to warrant the claim that the body-ownership effect is at the awareness level rather than the low-level perceptual or ocular dominance stage.

Perhaps this is what the authors intend by talking about how boosting signal strength can increase dominance times in binocular rivalry, and the other related purported neural mechanisms in the discussion (e.g. the functional connectivity between IPS and EBA is increased in "ownership" conditions). But I think the conceptual distinction needs to be made between a boost in internal signal strength that leads to increased awareness (longer dominance times), versus an effect at the higher "awareness" level itself. So I think the conclusion of the paper needs to be about how congruent visuo-tactile-proprioceptive integration leads to increased internal signal strength or more efficient processing, which can manifest as longer dominance times and higher "ownership" ratings, rather than that ownership leads to awareness itself.

I also have some comments about the design and analysis:

– Why the median split for the illusion scores (Q1 + Q2) – (Q3 + Q4)? Why not simply look at continuous measures (e.g. the Spearman correlation already presented)? The median split seems unnecessary.

– Despite the fact that the dominance time t-tests were planned, they should be corrected for multiple comparisons.

– I find it puzzling that the illusion is really only measured once for each participant. Why not assess after each block? It has been shown that the "no stimulation" condition used here also induces the illusion but to a lesser degree (Samad et al., 2015). So, the statement "ownership of the hand being absent during the other four conditions" is not necessarily accurate, but assumed throughout the study. This means the comparisons here might actually underestimate the effect of synchronous stroking on ownership and therefore dominance time, or might not adequately index the effect of ownership per se, but instead index the effect of ownership coupled with congruent visuotactile stimulation. Could the authors justify why the questionnaire was not used after each block? And why each condition (except the visuotactile condition) was only tested once per participant? This sparsity of data makes it difficult to assess the reliability of the results. This is especially interesting because here the no-stimulation condition (in Experiment 1) produced some similar but weaker effects to the congruent conditions; this is also related to the citation of Salomon et al's 2013 paper, which found congruency effects in CFS break times. I know that Experiment 2 somewhat addresses this, but overall I think the results would be stronger and more informative if ownership were more systematically queried.

– As a related point, the congruent visuo-tactile versus visual only condition is really a comparison of a trisensory congruent visuo-tactile-proprioceptive condition to a bisensory congruent (visuo-proprioceptive or tactile-proprioceptive) condition (Experiment 1), or to a trisensory incongruent visuo-tactile-proprioceptive condition (in which it is the proprioceptive information that is incongruent, Experiment 2). As a result, and because ownership was not systematically measured as much as I would have liked, it's hard to tell whether the results are due to *ownership per se* or simply more sensory information that happens to be congruent. The fact that there is increased effect with increased ownership in a between-subjects analysis suggests that ownership may be the cause, but this is somewhat confounded by the fact that there is simply more information in the trisensory congruent case, and it may be the congruency and not ownership that is driving the effect. Could the authors expand on this possibility, and how it relates to my previous point?

– I do not understand the normalization, described on in Materials and methods subsection “Duration of single percepts”. The authors state, "Thus, for hand- percepts and mask-percepts, we separately subtracted the individual mean percept duration across conditions and divided this difference by that same mean duration in the visual-only condition ((individual percept – mean)/mean)." So "mean" in this equation refers to two different things? In the numerator, it is the mean across all conditions, and in the denominator it is the mean in the visual-only condition? It would be helpful if the notation could be updated to more precisely reflect what has actually been done. In this same section, the authors also state, "… we collapsed the data from both experiments for the visuotactile and visual-only conditions (which were identical across experiments)." But this is not true: the stimulation was identical, but the conditions were not, because in Experiment 1 the hand was spatially congruent with the image, and in Experiment 2 the hand was spatially incongruent with the image. I would suggest this text be revised to more precisely specify the logic and analysis procedure.

Reviewer #3:

This is an interesting idea that was implemented in a clever way and certainly provides value to the field. That said, I have a few concerns that are detailed below.

For experiment 1, it is difficult to rule out the alternative explanation that the increase in dominance was due to the presence of visual motion (the main effect of visual was significant and the interaction was not). The movement of the visual stimulus perhaps makes those conditions that have it more salient, and thus that it is more likely to break through the mask and dominate. It is plausible that the increase in ownership in the visuotactile condition is just a result of this (i.e. more visuotactile integration merely as a result of more vision of the hand).

Subsection “Hand-ownership illusion” paragraph one: With only 19 out of the 30 giving a positive rating for the ownership questionnaire item that directly assesses ownership, and as shown in Figure 2 the interquartile range for Q1 is almost symmetric around 0, it is a stretch to claim that "on a group level, ownership of the hand was successfully induced". The fact that they rated illusion statements higher than control statements does little to convince me of this (it is not necessarily a fair comparison as the internal scales by which subjects are rating these questions may be very different as they are worded very differently). A better way to establish it would be to find a significant difference between the ownership question on this condition as compared with a condition that is not expected to induce ownership (as was used in Experiment 2). Therefore I would suggest the authors temper their claim that ownership was induced, perhaps by suggesting that there is evidence that it may have been induced.

Results subsection “Total dominance hand-percept” paragraph one: here you seem to report many more t-tests than what was described as the planned tests in the Materials and methods subsection “Total dominance hand-percept, paragraph two. There, the claim was that the planned t-tests were between conditions that used the identical visual stimuli, whereas the results report t-tests comparing the visuotactile condition vs all others. Would these tests all survive correction for multiple comparisons?

Subsection “The effect of individual touches on overall dominance”: What was the decision process involved in limiting the number of post-hoc tests to just the one between T0 and T2 that is reported? It seems to me that there are several other possible tests, and also that if these tests were conducted, they should be reported with the corresponding relevant Bonferroni correction.

Experiment 2 does a good job of mitigating the problem of the visual motion confound that was present in experiment 1. The significance of the interaction between congruence and visuotactile stimulation on dominance, coupled with the corresponding effect on ownership, provide good evidence that ownership is indeed driving the increase in dominance. However, there are still a few remaining issues that need to be addressed.

The scatterplot of dominance versus ownership for experiment 2 missing. It would be interesting to see that analysis as well as the correlation. If the main conclusion of the paper is trying to state that ownership causes an increase in dominance, this seems to be a critical way to show that.

Figure 9 and Figure 3 seem to be in conflict. In examination of the dominance as a function of time around the individual touches (Figure 9), I cannot see how the visual-only congruent condition can possibly be smaller on average than the two incongruent conditions, as Figure 3 seems to show. Does this imply an error in the data analysis? Or rather, does it indicate that this happens independently of the touches (i.e. at time points that are beyond the -667-1333 ms window examined in these time-courses)?

[Editors' note: further revisions were requested prior to acceptance, as described below.]

Thank you for resubmitting your work entitled "Body ownership promotes visual awareness" for further consideration at *eLife*. Your revised article has been favorably evaluated by Sabine Kastner (Senior and Reviewing editor), and two reviewers.

The manuscript has been improved but there are some remaining issues that need to be addressed before acceptance, as outlined below:

While both reviewers pointed out that they found the revisions overall satisfactory, they both had a few additional comments that need to be addressed, before we can make our final decision on publication.

Reviewer #2:

The authors have adequately addressed my concerns, for the most part, although some of my concerns (e.g. ownership assessed more frequently and more comprehensively; differentiating between "more information that is congruent" versus "ownership") cannot be addressed because the data are not there to support new analyses.

Regarding the authors' response to my Comment 1: I think I was not clear enough in my previous point, for which I apologize. The point really wasn't about the difference between perceptual awareness versus subjective awareness. It was about whether we can say anything about the effect of *ownership* on awareness per se, versus just signal strength (or "more information" that happens to be congruent) that ends up manifesting in the awareness measure. For example, we probably wouldn't claim that turning up the brightness on a computer screen or turning up the volume leads to more 'awareness' – it boosts signal strength which leads to more awareness, but if it's awareness itself that we're interested in, that's not quite the same thing. Likewise, boosting signal strength *internally* also doesn't boost *awareness* itself – but that boost in signal strength (or fidelity, like with attention) can *manifest* as more "seen" reports in an awareness task. But these really aren't the same thing, and it's not clear that ownership is what was doing the boosting anyway.

I do appreciate that the authors refer to Lamme's recurrent processing theory as support for their interpretation, but as they point out GWT (Dehaene) would not agree. Likewise, higher order theories certainly do not agree that a boost in signal strength = a boost in awareness.

I hope this will help clarify my concerns. I'm not asking the authors to differentiate between subjective versus perceptual awareness, but rather to acknowledge that in this project the measured 'awareness' might, in fact, be nothing more than a boost in signal strength at a much lower level of processing – akin to turning up the brightness on a computer monitor. This is also not to say that this finding is not *interesting*, but just that it's not the same as boosting awareness per se. Perhaps we are talking past each other here, but a little clarification about this in the manuscript itself, rather than just in the reply to reviews, would be beneficial to the reader.

Although as I said above the authors have done an acceptable job addressing my other concerns, this one is still at the forefront and really, in my mind, requires a change in the text itself rather than in the replies to me. It really calls for a toning-down of the interpretation of the results, and seems to alter the breadth of the conclusions the authors want to make.

Reviewer #3:

Thank you for addressing the majority of the comments raised and clearing up many of the confusions I had while reading this. There are still just a few outstanding issues:

Results subsection “Hand ownership illusion” still contains text describing the median score analysis even though the methods reporting that have been removed and replaced with the correlation analysis method. Please remove this to avoid confusion.

Thanks for adding the analysis and plot of the correlation between dominance and ownership for experiment 2. While not crucial from the point of view of this review, I would be curious to see what the correlation looks like for the incongruent visuotactile condition as well (at the very least I strongly recommend including a report of the correlation itself if not also including the plot). The Wilcoxon-signed rank test already demonstrates that the illusion score is smaller in that condition, but it is interesting to look into whether there was nevertheless a relationship between the illusion and dominance even in a condition that had reduced ownership.

---

## [Author Response]

*Reviewer #1:*
[…]*A few comments throughout could be toned down. For example, in the Introduction, "In the current study, we used binocular rivalry to answer this question." A lot of readers will take issue with this misplaced confidence. The study does not answer the question. It addresses the question; it provides evidence; some people will be convinced (especially by the very nice experiment 2) but others will need more data to convince them it isn't a different, correlated process that explains the results. For example, it is always possible that body ownership causes a slight increase in attention, and that the crucial factor is attention, however it is directed, whether by body ownership illusion or not.*

We adjusted this sentence. Updated text:

“In the current study, we used binocular rivalry to address this question.”

*The last paragraph of the paper also contains some overly confident sentences. It's very tricky to claim that you have absolutely shown a causal relationship. The results suggest it.*

We updated the paragraph: “Together, the results of this study suggest that ownership of a hand increases the visual awareness of that hand, both by increasing the strength of the hand image, as if it increases in contrast, and by overcoming suppression when the hand is being touched. These findings increase the understanding of the role of ownership in visual perception in that ownership seems to enhance visual processing of a body part that is ours because it is more relevant to us, especially when it is being touched by an external source.”

We also modified the last sentence of the Abstract: “Together, these results suggest that the sense of body-ownership promotes visual awareness.”

*Reviewer #2:*
[…]*Personally, I'm not convinced that binocular rivalry measures visual awareness per se. Zou, He, and Zhang (2016, PNAS) showed that even unconsciously processed conflicting visual inputs can lead to conscious perceptual rivalry, which suggests that rivalry reflects a lower-level alternation or oscillation of ocular dominance rather than subjective awareness per se (see Giles, Lau, and Odegaard, 2016 for discussion). The same could theoretically be going on in the present study, in that the fluctuation in signal is occurring lower than the "visual awareness" stage. In the parlance of hierarchical causal inference in multisensory integration, perhaps the pre-conscious processing ends up selecting the visual input that is congruent with the posterior probability of a common cause that the system arrives at (see Samad, Chung, and Shams 2015). That is, when the perceptual system assigns a high posterior probability of a common cause (high binding probability), the visual input that is congruent with that inference is boosted (via gain enhancement, or some other mechanism of signal strength enhancement) while the visual input that is incongruent with the inference (or irrelevant to it) is suppressed. This may also be compounded by the fact that binocular rivalry experiments kind of force the observer into a model selection or probability matching decision strategy, rather than a model averaging kind of framework (see Wozny et al., 2010 PLoS Comp Bio), because it's less easy to blend the inputs from the two eyes in a weighted averaging kind of way (although partial breakthrough does occur). So if this is what's happening, it's a low level selection of ocular dominance and signal strength boosting that may have little to do with awareness per se.*
[…]*Perhaps this is what the authors intend by talking about how boosting signal strength can increase dominance times in binocular rivalry, and the other related purported neural mechanisms in the discussion (e.g. the functional connectivity between IPS and EBA is increased in "ownership" conditions). But I think the conceptual distinction needs to be made between a boost in internal signal strength that leads to increased awareness (longer dominance times), versus an effect at the higher "awareness" level itself. So I think the conclusion of the paper needs to be about how congruent visuo-tactile-proprioceptive integration leads to increased internal signal strength or more efficient processing, which can manifest as longer dominance times and higher "ownership" ratings, rather than that ownership leads to awareness itself.*

We thank the reviewer for this interesting analysis. The reviewer is right to argue that binocular rivalry might be resolved at a perceptual awareness level rather than a subjective awareness level, as is the case in the study of Zhou, He and Zhang, 2016. However, the distinction between perceptual awareness and subjective awareness touches upon the question regarding what (visual) awareness really is. Different theories of (visual) awareness assume different ideas as to what visual awareness is and which brain processes are involved (for a recent review: Klink, Self, Lamme and Roelfsema, 2015). The increased functional connectivity between the IPS and the EBA that we propose in the Discussion would increase recurrent processing in the EBA, which, according to some authors, equals an increase in visual awareness (the recurrent processing theory, e.g., Lamme, 2006), whereas others would claim that a larger global network needs to be involved (the global neuronal workspace theory, e.g., Dehaene, Charles, King and Marti, 2014). However, we think it is beyond the scope of the current manuscript to discuss the nature of visual awareness and the possible neural mechanisms that underlie it. Moreover, we do not draw any conclusions specifically about subjective awareness over and above perceptual awareness (Giles, Lau and Odegaard, 2016), and our data cannot distinguish between these two types of awareness. Therefore, if possible, we would prefer not to speculate about this issue in the text.

Could the authors provide some citation for the statement, 'Because one's own body is more relevant than any other object'?

This sentence has been updated, as it is a logical statement, and not based on empirical evidence. We use our body to interact with the world, and we need to protect it from harm. Or to put it differently: we are our body. Therefore, it can be assumed to be more relevant than any other object. Updated text:

“Because one’s own body can be assumed to be more relevant than any other object it is reasonable to expect that visual awareness will be boosted when, all else being equal, body ownership is present.”

*I also have some comments about the design and analysis:*
*– Why the median split for the illusion scores (Q1 + Q2) – (Q3 + Q4)? Why not simply look at continuous measures (e.g. the Spearman correlation already presented)? The median split seems unnecessary.*

We agree with the reviewer that the comparison between the strong-ownership group and the weak-ownership group is redundant. We removed it from the updated manuscript.

*– Despite the fact that the dominance time t-tests were planned, they should be corrected for multiple comparisons.*

We planned the experiments to test one specific hypothesis: body ownership increases visual awareness of a hand image during binocular rivalry. The alternative hypothesis was that body ownership does not affect visual awareness during binocular rivalry. When testing a specific hypothesis, it is not logical to perform corrections for multiple comparisons (Perneger, 1998). There are several reasons for this claim. Assuming, for the sake of argument, that multiple testing corrections would always be a correct practice, this would imply that the interpretation of the data from a given study depends on the number of tests performed in the analysis and not on the information encoded in the data itself. Moreover, multiple-testing corrections, which have the main goal of reducing Type I errors, have the related drawback of inflating Type II errors, and this effect is problematic as well. In fact, if one specific hypothesis predicts several different effects of an experimental manipulation and each of these effects is found to be significant at the *p* < 0.05 level, correctly rejecting the null hypothesis would logically be more likely rather than less likely. For all these reasons, we did not plan to apply corrections for multiple comparisons when we designed the experiments, and we have therefore not included Bonferroni corrections for our planned comparisons in the current version of the manuscript. We do correct for multiple comparisons in our post hoc tests.

*– I find it puzzling that the illusion is really only measured once for each participant. Why not assess after each block? It has been shown that the "no stimulation" condition used here also induces the illusion but to a lesser degree (Samad et al., 2015). So, the statement "ownership of the hand being absent during the other four conditions" is not necessarily accurate, but assumed throughout the study. This means the comparisons here might actually underestimate the effect of synchronous stroking on ownership and therefore dominance time, or might not adequately index the effect of ownership per se, but instead index the effect of ownership coupled with congruent visuotactile stimulation. Could the authors justify why the questionnaire was not used after each block? And why each condition (except the visuotactile condition) was only tested once per participant?*

We agree with the reviewer that it would have been informative to measure ownership after each condition. However, there were two main reasons why we did not include a questionnaire after each block. First, we measured the illusion only once per participant in Experiment 1 and twice per participant in Experiment 2, and these subjective measures of ownership were taken after participants had completed all binocular rivalry trials. This design ensured that the binocular rivalry data were unaffected by participants’ knowledge of ownership illusions. Measuring ownership after each block, however, would allow for unwanted compliance and expectancy effects. Second, because the ownership questionnaire contains items that refer to tactile sensations and their relation to visually perceived movement (e.g., “it felt as if the touch I felt was caused by the white ball that I saw”), several conditions render these illusion statements nonsensical. For example, in the tactile-only condition there was no moving probe to be seen, and in the visual-only condition there was no tactile sensation.

We agree with the reviewer that possible weak sensations of ownership could have arisen in the no-stimulation condition (as in Samad et al., 2015), which, if anything, would reduce the described experimental effects. In the new version of the manuscript, we mention this possibility in the Materials and method section. However, it is important to note the very well-established fact that synchronous visuotactile stimulation elicits very strong sensations of limb ownership. In the study of Samad et al., 2015, the addition of synchronous strokes made the illusion of ownership stronger compared with the condition in which the participants merely looked at the model hand. Other studies reported that conditions in which participants merely looked at a hand did not elicit significant ownership illusions in the majority of the participants (Holmes et al., 2006; Guterstam et al., 2016; unpublished observations). Thus, we think it reasonable to assume that limb ownership in our no-stimulation condition should be significantly reduced compared with the synchronous visuotactile condition. Nevertheless, the reviewer is correct that we cannot fully exclude the possibility that the participants experienced (weak) ownership in the no-stimulation condition. Therefore, we have updated the following text:

“Experiment 1 consisted of five conditions, with ownership of the hand being induced during one condition (“visuotactile”) and ownership of the hand being absent or reduced during the other four conditions (see below).”

“On the basis of what is known from a large body of work on the rubber-hand illusion, we assumed that the visual-only and tactile-only conditions would not elicit a hand-ownership illusion because they contain a mismatch between vision and touch (Ehrsson, 2012). The no-stimulation condition could theoretically induce a weak ownership illusion since there was less sensory mismatch between different modalities, but such a putative illusion would be significantly weaker compared to the visuotactile condition (Samad, Chung, and Shams, 2015; Guterstam et al., 2016).”

*– As a related point, the congruent visuo-tactile versus visual only condition is really a comparison of a trisensory congruent visuo-tactile-proprioceptive condition to a bisensory congruent (visuo-proprioceptive or tactile-proprioceptive) condition (Experiment 1), or to a trisensory incongruent visuo-tactile-proprioceptive condition (in which it is the proprioceptive information that is incongruent, Experiment 2). As a result, and because ownership was not systematically measured as much as I would have liked, it's hard to tell whether the results are due to *ownership per se* or simply more sensory information that happens to be congruent. The fact that there is increased effect with increased ownership in a between-subjects analysis suggests that ownership may be the cause, but this is somewhat confounded by the fact that there is simply more information in the trisensory congruent case, and it may be the congruency and not ownership that is driving the effect. Could the authors expand on this possibility, and how it relates to my previous point?*

We agree with the reviewer that, in Experiment 1, the congruent visuotactile condition contained more information (trisensory) than any of the other conditions. Experiment 2 was designed to counteract this confounding factor, such that we could compare two conditions with the same amount of sensory information that differed significantly in experienced ownership.

We understand the point raised by the reviewer that it is difficult to dissociate the effect of ownership from that of ‘congruent trisensory information.’ However, in our view, ownership is the direct consequence of body-centered ‘congruent trisensory information’ (Ehrsson, 2012) and therefore it is theoretically very difficult to dissociate the two. Moreover, experimental manipulation of the spatial congruency of multisensory body-related signals is a commonly used approach to manipulate body ownership in the rubber hand illusion literature (e.g., Pavani et al., 2000; Ehrsson et al., 2004; Tsakiris and Haggard, 2005; Kalckert and Ehrsson, 2012).

*– I do not understand the normalization, described on in Materials and methods subsection “Duration of single percepts”. The authors state, "Thus, for hand- percepts and mask-percepts, we separately subtracted the individual mean percept duration across conditions and divided this difference by that same mean duration in the visual-only condition ((individual percept – mean)/mean)." So "mean" in this equation refers to two different things? In the numerator, it is the mean across all conditions, and in the denominator it is the mean in the visual-only condition? It would be helpful if the notation could be updated to more precisely reflect what has actually been done. In this same section, the authors also state, "… we collapsed the data from both experiments for the visuotactile and visual-only conditions (which were identical across experiments)." But this is not true: the stimulation was identical, but the conditions were not, because in Experiment 1 the hand was spatially congruent with the image, and in Experiment 2 the hand was spatially incongruent with the image. I would suggest this text be revised to more precisely specify the logic and analysis procedure.*

We thank the reviewer for pointing out the lack of clarity in section 2.5.4. The text has been updated to clarify the normalization procedures. To be clear, the visuotactile condition and the visual-only condition of Experiment 1 (in which the participants’ hand position was congruent to the hand image) are identical to the congruent visuotactile condition and the congruent visual-only condition of Experiment 2 (which is reflected by the color-coding of the figures in the Results section). Updated text:

“Thus, separately for hand-percepts and mask-percepts, we subtracted the participants’ mean percept duration across all conditions from that of individual conditions and divided this difference by that same mean ((condition-mean –mean)/mean). […] The other conditions were not identical and therefore we re-normalized the data by dividing by the mean of only the visuotactile condition and the visual-only condition; ((visuotactile – visual-only) / mean) (mean = 0.5*(visuotactile + visual-only)).”

*Reviewer #3:*

*This is an interesting idea that was implemented in a clever way and certainly provides value to the field. That said, I have a few concerns that are detailed below.*
*For experiment 1, it is difficult to rule out the alternative explanation that the increase in dominance was due to the presence of visual motion (the main effect of visual was significant and the interaction was not). The movement of the visual stimulus perhaps makes those conditions that have it more salient, and thus that it is more likely to break through the mask and dominate. It is plausible that the increase in ownership in the visuotactile condition is just a result of this (i.e. more visuotactile integration merely as a result of more vision of the hand).*

The main effect of the visual component of touch is indeed very strong. It is important to keep in mind that this visual motion was present in both images (the hand-image and the mask-image). This suggests that the hand provides a more congruent context for the moving object than the mask does (see Discussion). In addition to the effect of motion, the effect of congruent, synchronous tactile stimulation in the visuotactile condition increases dominance of the hand further. The difference between the visuotactile condition and the visual-only condition shows that the dominance increase is caused by something other than the moving object.

Although it is theoretically possible that an increase in ownership in the visuotactile condition is a result of increased visibility of the hand image, we did not find any evidence for such a relationship. We found that the strength of subjective ownership correlated with the increasein hand-image dominance (from participants’ individual mean). The absolutehand dominance (*i.e*., the total time the hand image was seen) during the visuotactile condition did not correlate with illusion score (Experiment 1: *ρS* = -0.03, *p* = 0.87; Experiment 2: *ρS* = 0.19, *p* = 0.31).

*Subsection “Hand-ownership illusion” paragraph one: With only 19 out of the 30 giving a positive rating for the ownership questionnaire item that directly assesses ownership, and as shown in Figure 2 the interquartile range for Q1 is almost symmetric around 0, it is a stretch to claim that "on a group level, ownership of the hand was successfully induced". The fact that they rated illusion statements higher than control statements does little to convince me of this (it is not necessarily a fair comparison as the internal scales by which subjects are rating these questions may be very different as they are worded very differently). A better way to establish it would be to find a significant difference between the ownership question on this condition as compared with a condition that is not expected to induce ownership (as was used in Experiment 2). Therefore I would suggest the authors temper their claim that ownership was induced, perhaps by suggesting that there is evidence that it may have been induced.*

We agree with the reviewer that comparing the same statement across different conditions is preferred over comparing individual statements within a condition. The reason for not including an ownership questionnaire in the other conditions is discussed above in our reply to reviewer 2.

The finding that 2/3 of the participants in Experiment 1 affirmed statement Q1 (“it felt as if the hand I saw was my own hand”) is very similar to the findings of previous studies that investigate ownership during the rubber hand illusion (Kalckert and Ehrsson, 2014). In fact, it is notable that the subjectively reported ownership is comparable to that of the traditional rubber hand illusion despite the fact that the hand image was only viewed by one eye and that it was not visible at all times. Nevertheless, we have toned down the conclusion in line with the reviewer’s suggestion and explicitly comment upon the fact that the questionnaire results are comparable to the results of previous rubber hand illusion experiments. A similar update has been made to the Discussion.

Updated text:

“Nineteen participants out of the total group of 30 rated at least +1 on questionnaire statement Q1 (“It felt as if the hand I saw was my own hand”), and the median response was +1 (Figure 2). In addition, 23 participants rated at least +1 on Q2 (“The touch I felt seemed to be caused by the white ball I saw”), and the median was +2. These proportions of participants affirming the illusion related statements and the median scores are comparable to previous experiments with the rubber hand illusion (Kalckert and Ehrsson 2014). The median rating of the control statements was lower than zero (Q3: Mdn = -1; Q4: Mdn = -2.5). Crucially, participants rated the illusion statements significantly higher than the control statements (Z = 4.44, *p* < 0.001, r = 0.57). Thus, on a group level, these questionnaire results suggest ownership of the hand was successfully induced during the visuotactile condition.”

and:

“However, the results of our questionnaire data demonstrated that we elicited a relatively strong limb-ownership illusion, with affirmative ratings of ownership that were comparable to those from previous studies with actual rubber hands (Kalckert and Ehrsson, 2014).”

*Results subsection “Total dominance hand-percept” paragraph one: here you seem to report many more t-tests than what was described as the planned tests in the Materials and methods subsection “Total dominance hand-percept, paragraph two. There, the claim was that the planned t-tests were between conditions that used the identical visual stimuli, whereas the results report t-tests comparing the visuotactile condition vs all others. Would these tests all survive correction for multiple comparisons?*

We removed this inconsistency. We now clearly state which comparisons were planned and which ones were post hoc. The post hoc tests are corrected for multiple comparisons. Updated text:

“Planned comparison revealed that the overall dominance of the hand was significantly larger in the visuotactile condition compared to the visual only condition (*mean difference* ± *SEM*= 4.48 ± 1.76, *CI* = 0.87 – 8.08, *t*(28) = 2.54, *p* = 0.017, *d* = 0.47). Additional post hoc tests showed significantly larger dominance in the visuotactile condition compared to the three other conditions as well (visuotactile vs baseline: *mean difference* ± *SEM*= 9.70 ± 2.36, *CI* = 4.88 – 14.53, *t*(28) = 4.11, *p* = 0.003, *d* = 0.75; vs no-stimulation: *mean difference* ± *SEM*= 9.10 ± 1.99, *CI* = 5.03 – 13.16 *d* = 0.85, *t*(28) = 4.58, *p* = 0.002; vs tactile-only: *mean difference* ± *SEM*= 7.07 ± 1.61, *CI* = 3.78 – 10.36, *t*(28) = 4.40, *p* < 0.001, *d* = 0.80; vs visual-only) (Figure 3).”

*Subsection “The effect of individual touches on overall dominance”: What was the decision process involved in limiting the number of post-hoc tests to just the one between T0 and T2 that is reported? It seems to me that there are several other possible tests, and also that if these tests were conducted, they should be reported with the corresponding relevant Bonferroni correction.*

We included a new table (Table 1) to show all t-tests that were used to assess during which time interval the dominance started increasing from the onset of touch. These statistics are Bonferroni corrected.

*Experiment 2 does a good job of mitigating the problem of the visual motion confound that was present in experiment 1. The significance of the interaction between congruence and visuotactile stimulation on dominance, coupled with the corresponding effect on ownership, provide good evidence that ownership is indeed driving the increase in dominance. However, there are still a few remaining issues that need to be addressed.*
*The scatterplot of dominance versus ownership for experiment 2 missing. It would be interesting to see that analysis as well as the correlation. If the main conclusion of the paper is trying to state that ownership causes an increase in dominance, this seems to be a critical way to show that.*

We agree with the reviewer that the correlation between ownership and the perceptual effect of Experiment 2 is an interesting analysis to include in the manuscript. We included this correlation and an associated figure (Figure 4) in the updated manuscript.

New text:

“In a post hoc correlation analysis, we found that, similar to Experiment 1, participants’ ownership illusion scores correlated positively with the increased dominance of the hand-image in the congruent visuotactile condition (*ρS* = 0.64, *p* < 0.001) (Figure 4). This finding provides further evidence that ownership of the observed hand drives the increase in dominance of the hand-image.”

and:

“Crucially, in both experiments, we found that this dominance increase correlated positively with participants’ individual ownership illusion score.”

Figure 9 and Figure 3 seem to be in conflict. In examination of the dominance as a function of time around the individual touches (Figure 9), I cannot see how the visual-only congruent condition can possibly be smaller on average than the two incongruent conditions, as Figure 3 seems to show. Does this imply an error in the data analysis? Or rather, does it indicate that this happens independently of the touches (i.e. at time points that are beyond the -667-1333 ms window examined in these time-courses)?

The latter is indeed the case. The analyses on the effect of individual touches use only the data around single touches (-667 – 1334 ms). As explained in the Method section, the pattern of touches was as follows: three strokes – one rest – two strokes – one rest. During the ‘rest’, the tactile probe came to a full stop next to the hand, and the dominance of the hand image dropped considerably, perhaps as a rebound effect from the stimulating effect of the movement.

[Editors' note: further revisions were requested prior to acceptance, as described below.]

*Reviewer #2:*

*The authors have adequately addressed my concerns, for the most part, although some of my concerns (e.g. ownership assessed more frequently and more comprehensively; differentiating between "more information that is congruent" versus "ownership") cannot be addressed because the data are not there to support new analyses.*
*Regarding the authors' response to my Comment 1: I think I was not clear enough in my previous point, for which I apologize. The point really wasn't about the difference between perceptual awareness versus subjective awareness. It was about whether we can say anything about the effect of *ownership* on awareness per se, versus just signal strength (or "more information" that happens to be congruent) that ends up manifesting in the awareness measure. For example, we probably wouldn't claim that turning up the brightness on a computer screen or turning up the volume leads to more 'awareness' – it boosts signal strength which leads to more awareness, but if it's awareness itself that we're interested in, that's not quite the same thing. Likewise, boosting signal strength *internally* also doesn't boost *awareness* itself – but that boost in signal strength (or fidelity, like with attention) can *manifest* as more "seen" reports in an awareness task. But these really aren't the same thing, and it's not clear that ownership is what was doing the boosting anyway.*
*I do appreciate that the authors refer to Lamme's recurrent processing theory as support for their interpretation, but as they point out GWT (Dehaene) would not agree. Likewise, higher order theories certainly do not agree that a boost in signal strength = a boost in awareness.*
*I hope this will help clarify my concerns. I'm not asking the authors to differentiate between subjective versus perceptual awareness, but rather to acknowledge that in this project the measured 'awareness' might, in fact, be nothing more than a boost in signal strength at a much lower level of processing – akin to turning up the brightness on a computer monitor. This is also not to say that this finding is not *interesting*, but just that it's not the same as boosting awareness per se. Perhaps we are talking past each other here, but a little clarification about this in the manuscript itself, rather than just in the reply to reviews, would be beneficial to the reader.*

We thank Reviewer 2 for clarifying this comment. We fully agree that the increase in perceptual dominance of the hand image could possibly be due to increased signal strength of that image, which affects the perceptual awareness only “indirectly”. Such an increase in signal strength would nonetheless be mediated by the multimodal integration that comprises ownership since visual input was controlled for. Alternatively, the effect of ownership might “directly” influence by enhancing recurrent processing from the IPS to the EBA. In a new discussion paragraph we outline these two possibilities and explicitly discuss the conceptual distinction raised by the reviewer.

New text:

“The above discussion brings us to the conceptual issue of whether body ownership boosts visual awareness “directly” at the higher level of conscious selection, or if body ownership influences visual awareness “indirectly” by first increasing the signal strength in lower level visual representations, which in turn leads to facilitation of awareness (Giles, Lau, and Odegaard, 2016). […] Thus, although our results clearly show that body ownership increases the subjective dominance of the hand image, the possible existence of a direct causal relationship between body ownership and conscious awareness remains to be clarified in future studies.”

*Reviewer #3:*
*Thank you for addressing the majority of the comments raised and clearing up many of the confusions I had while reading this. There are still just a few outstanding issues:*
*Results subsection “Hand ownership illusion” still contains text describing the median score analysis even though the methods reporting that have been removed and replaced with the correlation analysis method. Please remove this to avoid confusion.*

We thank the reviewer for pointing this out. This text has been removed in the updated manuscript.

*Thanks for adding the analysis and plot of the correlation between dominance and ownership for experiment 2.*

We would like to make a correction to the correlation between perceptual dominance and ownership of Experiment 2. Although the supplied figure (Figure 4) is correct, an error occurred in the statistical analysis. The correlation did not reach significance (*ρ_S_* = 0.24, *p* = 0.197). Updated text:

“In a post hoc correlation analysis, we found that, in contrast to Experiment 1, the positive correlation between participants’ ownership illusion scores and the increased dominance of the hand-image in the congruent visuotactile condition did not reach significance (*ρ_S_* = 0.24, *p* = 0.197) (Figure 4).”

To be clear, the non-significant correlation in Experiment 2 does not affect our conclusions. This experiment was designed to directly manipulate ownership through manipulating spatial congruency, and this manipulation had significant effects on perceptual dominance of the hand image as we had predicted. Thus our conclusion that ownership of the hand image promotes the perceptual dominance of the hand is supported by the results. Not finding a significant correlation in an explorative post-hoc test does not falsify this conclusion. In our opinion we could take out the correlation analysis from the manuscript and our main conclusions would not change.

Furthermore, it is important to note that the perceptual dominance in the congruent visuotactile condition is normalized based on different control conditions in Experiment 1 and Experiment 2. As a result, the normalized effect is on average smaller in Experiment 2 (mean normalized dominance in visuotactile condition of Experiment 1 = 0.182; mean normalized dominance in congruent visuotactile condition in Experiment 2 = 0.064), and therefore there is *relatively* more noise in the normalized perceptual dominances of Experiment 2, which might partially explain the lack of a significant correlation.

*While not crucial from the point of view of this review, I would be curious to see what the correlation looks like for the incongruent visuotactile condition as well (at the very least I strongly recommend including a report of the correlation itself if not also including the plot). The Wilcoxon-signed rank test already demonstrates that the illusion score is smaller in that condition, but it is interesting to look into whether there was nevertheless a relationship between the illusion and dominance even in a condition that had reduced ownership.*

We conducted this correlation analysis, and found a similar trend for the incongruent visuotactile condition (*ρ_S_* = 0.23, *p* = 0.221) as we found for the congruent visuotactile condition (see Comment 2), with the difference that overall, ownership scores and perceptual dominance were lower in the incongruent condition. The figure below shows the correlation between illusion score and perceptual dominance for the incongruent condition (note: the scales on the axes differ from Figure 4). Given that the correlation did not reach statistical significance we did not include this result in the main text.

Holmes, Snijders & Spence (2006). Reaching with alien limbs: Visual exposure to prosthetic hands in a mirror biases proprioception without accompanying illusions of ownership. *Perception & Psychophysics*, *68*(4), 685-701.

Klink, Self, Lamme & Roelfsema (2015). Theories and methods in the scientific study of consciousness. In: The Constitution of Phenomenal Consciousness, Ed. Miller, 17-47.

Pavani, Spence & Driver (2000). Visual capture of touch: Out-of-the-body experiences with rubber gloves. *Psychological science*, *11*(5), 353-359.

Perneger (1998). What is wrong with Bonferroni adjustments. *British Medical Journal*, *316*(7139), 1236–1238.